# Modeling Litter Stocks in Planted Forests of Northern Mexico

Felipa de Jesús Rodríguez-Flores [1], José-Guadalupe Colín [2], José de Jesús Graciano-Luna [2] and José Návar [3,*]

1 Ingeniería en Tecnología Ambiental, Universidad Politécnica de Durango, Carretera Durango-México Km 9.5, Localidad Dolores Hidalgo, Durango C.P. 34300, Mexico; jesu_rgz@hotmail.com

2 Tecnológico Nacional de México/Instituto Tecnológico de El Salto, Calle Tecnológico No 101, Colonia La Forestal, El Salto, Durango C.P. 34942, Mexico; Josecolin8@gmail.com (J.-G.C.); gracluna@hotmail.com (J.d.J.G.-L.)

3 Tecnológico Nacional de México/Instituto Tecnológico de Ciudad Victoria, Blvd Emilio Portes Gil No 1301 Pte., Cd Victoria C.P. 87010, Tamaulipas, Mexico

* Correspondence: jose.navar@itvictoria.edu.mx; Tel./Fax: +52-81-12397599

**Abstract:** Litter, *LS*, is the organic material in which locates in the top A soil horizon, playing key ecological roles in forests. Models, in contrast to common allocation factors, must be used in *LS* assessments as they are currently absent in the scientific literature. Its evaluation assess the mass, input and flux of several bio-geo-chemicals, rainfall interception as one component of the local hydrology, and wildfire regimes, among others, hence its importance in forestry. The aim of this study was to: (i) develop models to assess *LS*, accumulation, and loss rates; and (ii) assess rainfall interception and fire regimes in 133 northern forest plantations of Mexico. Two developed techniques: the statistical model ($SM_{LS}$) and the mass balance budget model ($MBM_{LS}$) tested and validated local and regional *LS* datasets. Models use basal area, timber, aboveground tree biomass, litter fall, accumulation, and loss sub-models. The best fitting model was used to predict rainfall interception and fire behavior in forest plantations. Results showed the $SM_{LS}$ model predicted and validated *LS* datasets ($p = 0.0001$; $r^2 = 0.82$ and $p = 0.0001$; $r^2 = 0.79$) better than the $MBM_{LS}$ model ($p = 0.0001$; $r^2 = 0.32$ and $p = 0.0001$; $r^2 = 0.66$) but the later followed well tendencies of Mexican and World datasets; counts for inputs, stocks, and losses from all processes and revealed decomposition loss may explain ≈40% of the total *LS* variance. $SM_{LS}$ predicted forest plantations growing in high productivity 40-year-old stands accumulate $LS > 30$ Mg ha$^{-1}$ shifting to the new high-severity wildfire regime and intercepting ≈15% of the annual rainfall. $SM_{LS}$ is preliminarily recommended for *LS* assessments and predicts the need of *LS* management in forest plantations (>40-year-old) to reduce rainfall interception as well as the risk of high-severity wildfires. The novel, flexible, simple, contrasting and consistent modeling approaches is a piece of scientific information required in forest management.

**Keywords:** litter stock & accumulation rates; litter losses; model predictions; mass balance budget model; forest wildfires

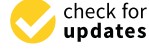



## 1. Introduction

The soil organic litter component of forests consists of fine litter composed mainly of leaf and wood litter [1]. Litter plays key roles in the ecology of forests such as regulating water cycles and as fuels of wildfire. Hence the need for measuring, developing and validating models to predict *LS* over time of forest growth. These mathematical techniques would help to objectively propose silvicultural practices that aim to optimize ecological functions while at the same time improving the soil water balance to increase productivity and reduce the risk of high-severity wildfires. Specifically, soil organic litters is crucial in ecosystem water cycles in its potential impact on wildfire. Litters are strongly correlated to the amount of Above Ground Biomass (AGB), so studies of this correlation can provide a better understanding of the water budget and the risk of high-severity wildfires. However predicting models cannot be found in a brief scientific literature. Common constant

allocation factors had been conventionally employed in *LS* assessments [2]. Forest growth and yield models aid in the evaluation of timber growth [3] and biomass expansion factors, BEF, [4] can aid in the evaluation of AGB as the major input of *LS* by developing empirical equations. Whole stand timber growth and yield models are based on evaluating stand variables such as basal area, BA, productivity indicators such as site index [5] and age of forests [6]. These variables can be used as input exogenous predictors to more complex statistical and budget techniques of *LS* evaluations. Once *LS* had been predicted, its role in the local hydrology as well as on the wildfire regime can be used to recommend forest management practices aiming to improve the local water balance and to reduce the risk of high severity wildfires.

Litter plays key roles in the hydrology of forests because it (i) intercepts from 2 to 70% of the annual rainfall [7–12] and (ii) reduces soil water evaporation [13–17]. The former is conventionally considered a water redistribution as intercepted rainfall seems to contribute little to soil moisture renewal, aquifer recharge or streamflow generation but maintain soil moisture and reduce soil erosion [18]. The latter signifies savings as litter shields the soil from climate reducing considerably the evaporation of soil water. The final water balance has profound implications on the hydrology of forests that has been recently modeled [19]. Two major components are considered as the key element for litter-mediated changes in water cycling: water holding capacity, $\theta_L$, and depth of rainfall intercept, $I_L$, has been recognized as important elements in litter interception [20]. While $\theta_L$ capacity is often larger than depth of rainfall $I_L$, $\theta_L$ could limit $I_L$ of the following rain; those two factors use to determine litter stocks [7,20]. Unfortunately litter mass balance models have not yet been proposed as an aid to simulate more precisely the role of litter on the hydrology of growing forests.

The accumulation of litter on forest soils represents one of the main fuel sources for sustaining ground wildfires [21]. For instance, several coniferous forests of Durango, Mexico hold a mean value of 50 Mg ha$^{-1}$ and the lag time since the last harvest partially explained the amount of litter stocks [22]. Coniferous forest stands of Chihuahua, Mexico had an average of 40 Mg ha$^{-1}$; here, too, the lag time since the last prescribed fire accounted for the variance of litter fuels [18]. The frequency of high-severity wildfires appear to be accelerating in World forests and northern Mexico has had at least four major wildfire seasons (1998, 2000, 2011, 2017) in the last 25 years [23]. Due in part to the present-day increased frequency of high-severity wildfires litter management is becoming a common practice. As a consequence, predictions of litter stocks are key to develop fire risk based on litter stocks, or which forest stands require litter management practices including prescribed burning, litter collection, litter arrangement, and redistribution. The fuel model for estimating fire behavior [24] can be fitted to total measured litter stocks or litter stocks projected with derived models. This procedure allows forecasting the time required by forests to shift to a risk factor of high-severity wildfires.

The litter mass balance is made of inputs (gains), outputs (losses), and changes in storage. Litter fall is the main input for litter stocks with average statistics varying in the range of 3–11 Mg ha$^{-1}$ y$^{-1}$ for world forests. The marked spatial and temporal variability can be explained by physiological, forest, and environmental variables [25]. Decomposition, wildfires, and surface runoff are the main processes contributing to litter losses. Decomposition of litter refers to the physical and chemical processes involved in reducing litter to its chemical constituents [26]. Current models predict decomposition rates and remaining litter stocks in the form of organic carbon (C) and nitrogen (N). Long-term field experiments such as CIDET, DECO, LIDET, and other studies using litterbag data are often employed for model fitting and validation [27]. Most litterbag-based studies typically examine decay rates and remaining pool in undisturbed forest conditions. Litterbag studies have examined decay rates under different stand conditions, microenvironments, and disturbed ecosystems [28].

Wildfires represent also a major loss of litter in forests. For example, on average, in the Appalachian forests of USA, high-severity wildfires may reduce the mass of litter stocks

from 180 Mg ha$^{-1}$ to 70 Mg ha$^{-1}$ [29] and worldwide statistics show high-severity, large-scale forest wildfires can burn a mean of 140 Mg ha$^{-1}$ of the total litter stocks [30,31]. Large variations in the mass of litter stocks and losses have been reported and explained as a function of the amount, composition (leaves vs. woody debris; type of litter), accumulation rate, and moisture content of litter that play key roles in the frequency, spreading rate, scale, intensity, and magnitude of wildfires [32]. Inclusion of other causes of litter loss, such as surface runoff, and harvest, requires the tuning of ecosystem models (e.g. CBM-CFS2, ECOSYS, and LANDIS) which currently do not fully account for such agents. In fact, the relative importance of different causes of litter loss in forests has received little attention in the scientific literature and at this time models that forecast all losses together must be developed. Meanwhile better mathematical descriptions by decomposition, wildfires, runoff are individually proposed.

In light of this brief literature review, the objectives of this study were to: (i) develop models to assess litter stocks, accumulation, and loss rates; and (ii) evaluate litter rainfall interception and the litter accumulation time when shifting to a high-severity fire regime in 133 northern forest plantations of Mexico. Models of litter stocks, accumulation rates, and losses are absent in the scientific literature and they are needed in order to better manage this forest component so as to meet several objectives such as the improvement of the water balance and the forecasting of the risk of high-severity wildfires. The central hypothesis of this study was that fitted models would statistically explain at least part of the large spatial variability expected in the mass of litter stocks measured in three different Mexican States (Durango, Coahuila and Nuevo Leon). Measured forest plantations have different basal area or stand density at the time of planting and at the time of measurements. Plantations with different pine species may also present different litter fall rates, decomposition factors, among many other traits.

## 2. Materials and Methods

### 2.1. Study Area

This research was conducted in non-industrial tree plantations within the western and eastern Sierra Madre mountain ranges in northern Mexico (Figure 1).

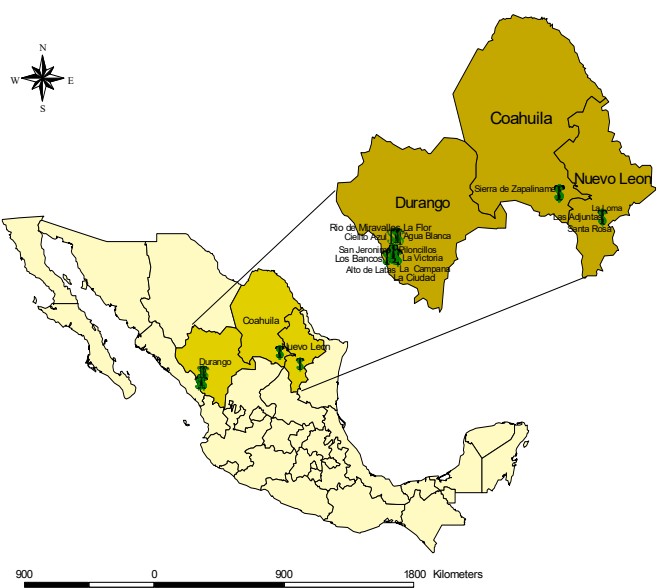

**Figure 1.** Location of northern Mexico non-industrial planted forest stands sampled for litter stocks.

In the states of Durango, Nuevo Leon, and Coahuila, these ranges are characterized by cold-temperate, dry-temperate, and dry climates, respectively. Mean annual temperatures and precipitation for the regions in which the planted plots are located are 14 °C and 950 mm, 18 °C and 450 mm, and 15 °C and 600 mm for Durango, Nuevo Leon, and

Coahuila, respectively. In Durango, native pine forests cover close to 4 M ha while in Coahuila and Nuevo Leon native forest cover is less than 0.3 M ha in each state [33]. In all three areas, pine tree species are distributed over an altitude range of 1600–3000 meters above sea level.

Reforestation has been carried out in these mountain ranges to restore plant cover in forests deprived of native vegetation by wildfires, pests and diseases, and degraded soils related to overharvesting and land degradation practices. Tree plantings in these areas are conducted in plots of different sizes, planting densities, and usually with single pine species; while they tend to cover a perturbed area, most frequently plantations are restricted to specific locations within forest stand. In upland Durango, the most commonly planted pine species (*Pinus cooperi* C.E. Blanco, *Pinus durangensis* Martinez, and *Pinus engelmanii* Carriere) are native to the sites, but in several places *Pinus arizonica* Engelmann has also been planted. This species preferably colonizes upland forests farther north than the planted sites and were planted because of a lack of seedlings of the native local species. In upland Nuevo Leon, the preferred pine species for planting is the native *Pinus pseudostrobus* Lindl, while pinyon pine species (*Pinus pinceana* Gordon, *Pinus nelsonii* Shaw, and *Pinus cembroides* Zucc.) have been also planted to recover forests in the lower degraded ranges. These pinyon pines are not native to the planted sites in Nuevo Leon, and *Pinus pinceana* Gordon and *Pinus nelsonii* Shaw are listed in the rare and endangered species list [34]. *Pinus halepensis* Mill., an exotic pinyon pine species, has been extensively planted in upland sites near Saltillo in Coahuila. Planting densities vary between plots but distances of 1 m × 1 m (10,000 seedlings per ha) were most frequently found in this research.

### 2.2. Sampling

Forest plantations were sampled for litter stocks and tree dimensions in a field inventory during 2002–2003, selecting a chronosequence of plantations with different ages, from 1 to 44 with average of 16 years old, of 133 forest plantations following a randomly-stratified design. A representative set of plantations was chosen with different dimensions assuming planting times were different. Single sampling plots (20 m × 20 m) were established in each of 47, 46, and 40 plantation plots scattered across the forests of Durango, Nuevo Leon, and Coahuila, respectively. Additional measurements were made in 2014 in 9 previously-sampled plots in Durango to validate models.

### 2.3. Measurements

Forestry parameters as exogenous variables help to predict litter stocks using either empirical or mass balance budget models. A forest inventory was carried out on each plot of each plantation, measuring the diameter at breast height (DBH), basal diameter (Db), tree top height (Ht) and canopy cover (Cc) of all trees taller than 1.3 m. The measurements shown in Table 1 are plot-level variables covering the values for all trees measured within the plots. The plantation age was shared by all trees in the plot and measured plot, determined by a ring count of cores extracted from three randomly selected trees using an increment borer at breast height (1.3 m). To determine the age of the plantation, two years were added to the final ring count to consider the time needed for the trees to reach 1.3 m after planting. This seems reasonable according to the height growth and yield models developed by [35].

All litter material was collected in each sampled plot from three randomly selected 1 m × 1 m sub-plot, including the organic litter material (OL), femic material: OF + hemic OH, branches and fallen trees within the plot for fresh and dry-weight measurements. Total litter mass was calculated using 1 kg subsamples (including all foliage, fine, medium, and coarse woody debris) by multiplying the ratio of oven dry weight (oven dry weight is defined when the litter sample reached constant weight; from 24 to 30 hours of consecutive oven drying at 75 °C of constant temperature) to fresh weight by the total fresh weight measured in the field. We defined surface litter as the amount of all foliage and undecomposed litter remaining atop the A horizon. Small and coarse woody debris (e.g.,

branches > 5 cm diameter) and fallen trees were measured although they were a few in number, quite conspicuous, and showed high spatial variability on the sampled sites. The mass of both coarse and fine woody debris was included to determine total litter mass. The initial planting density was evaluated from the distance of the nearest neighbor trees for several trees with similar dimensions and the final planting density was calculated as the number of tress per ha at the time of measurement.

*2.4. Modeling Approaches*

Two different models were developed to evaluate the mass of litter stocks: a statistical model using stepwise procedures, $SM_{LS}$, and a mass balance budget model using litter inputs (gains), changes in storage (present litter stock) and calibrating it for litter outputs or losses, $MBM_{LS}$. Stepwise procedure and most regression analysis were conducted in SAS v 8.1 [36].

Litter stock models were developed only using the data collected during 2002–2003, with the data from the nine plots re-measured in 2014 used for model validation. The $SM_{LS}$ model used the measured plot variables initial and final density, average diameter, top height, canopy cover, basal area, timber volume and aboveground biomass. Selected variables served as input to a growth and yield model to determine timber volume and aboveground biomass. The $MBM_{LS}$ model used litter fall as inputs, evaluated litter outputs as litter losses and litter stocks as changes in storage. First, AGB was evaluated and used as input to a sigmoid equation to determine the annual rate of litter fall to evaluate cumulative litter fall as input to the mass balance model, in which litter stock outputs were assessed as the difference between assessments of cumulative litter fall inputs and litter stocks measured in this study. The sigmoid equation parameters were derived using previously published litter fall data for a chronosequence of planted plots in northern Mexico [37] as well as by plotting additional litter fall data sets collected for pine forests in the region, in Mexico as well as in World forests.

2.4.1. Plot-Level Variables

The plot-level variable mean basal area (BA) was calculated as follows:

$$BA = \left[ K(1 - e(-\beta_1 t))^{\beta_2} \right] \tag{1}$$

where BA = stand basal area ($m^2\ ha^{-1}$); K = steady state final basal area, and $\beta_1$, $\beta_2$ = statistical parameters. Note that BA is evaluated using Db. In order to assess BA with DBH a close approximation would be to divide basal area by a correction factor of 1.5 determined from trees where both Db and DBH measured. The transformation of BA calculated with DBH is reported throughout this paper.

Equation (1) slightly underestimated basal area for the Durango plantations when fitted to all forest stands including the Nuevo Leon and Coahuila plantations. Therefore, to be consistent with other BA models for native forests of Durango e.g., [38], the Chapman-Richards equation was forced to reach 50 $m^2\ ha^{-1}$ at 80 year old plantations using only plantation age and site index as exogenous variables. That is K ≈ 50 $m^2\ ha^{-1}$ for 80-year-old plantations.

Site Index

Site index, SI, is an indirect productivity evaluation indicator that uses the top height and age of dominant trees in the stand. Following the methodology proposed earlier [39], site index assessments and derivation using the Chapman-Richards model is depicted in Equation (2).

$$H_t = \beta_0(1 \pm \exp(-\beta_1 t))^{\beta_2} := SI15 = H\left(\frac{1 - \exp(-\beta_1 t_0)}{1 - \exp(-\beta_1 t)}\right)^{\beta_2} \tag{2}$$

where $H_t$ = tree top height of dominant trees, $t_0$ = index at age of 15 years, $t$ = current age (years), and $\beta_0$, $\beta_1$, and $\beta_2$ = model parameters.

The evaluation of site index is conventionally carried out on a species-by-species basis. Because of a lack of large chronosequences spreading over the full age range of a single pine species, the methodology was conducted for all species combined. Although Carmean et al. [5] showed that the site index approach works better for single species than for all species combined, the approach used in here for all species accounted for a large proportion of the height-age relationship. Site index is a composite variable that describes tree genetic and environmental variability to estimate the productivity of planted stands; and classifying plots for differential basal areas, timber volumes, timber biomass.

A second estimator of site productivity, which is a dryness parameter, was also tested in this research. It is the ratio of mean annual precipitation, *P*, to mean annual potential evapotranspiration, *Et*; *P*/*Et*. *P* and *Et* values were acquired from [40] for the climatic station of El Salto, Durango placed at 2550 meters above sea level and an altitude gradient accounted for differences in *P* and *Et* for each forest plantation. The equations to calculate *P* and *Et* gradients are described in (3) and (4), respectively:

$$P_i = 1000 + \frac{(A_i - 2550)}{100} \cdot Fcp \tag{3}$$

$$Et_i = 616 + \frac{(A_i - 2550)}{100} \cdot Fcet \tag{4}$$

where *Ai* = altitude above sea level of the forest plantation *i*; 2550 = altitude above sea level of the climatic station of El Salto, P.N., Durango, Mexico; *Fcp*, *Fcet* = Precipitation and evapotranspiration correction factors accounting for slope and aspect.

### 2.4.2. Statistical Model SM$_{LS}$

The SM$_{LS}$ is an empirical regression equation developed in stepwise procedures in Proc Reg [36] used to predict absolute values for litter stocks from a series of single equations for basal area, timber volume, aboveground biomass, using time and productivity of forest plantations. This model does not include estimates of litter fall or litter loss; it simply relates the mass of litter stock to plot level variables as shown in Equation (5):

$$\text{SM}_{LS} = f\left(\text{AGB, SI, }\left(\frac{P}{Et}\right)\right);\ AGB = f(V);\ V = f(\text{BA});\ \text{BA} = f(t) \tag{5}$$

where SM$_{LS}$ = Statistical model to evaluate litter stocks and accumulation rates (Mg ha$^{-1}$), *V* = timber volume (m$^3$ ha$^{-1}$), BA = basal area (m$^2$ ha$^{-1}$), SI15 = site index, *t* = age of plantations, and *P*/*Et* = dryness index.

### 2.4.3. Mass Balance Model MBM$_{LS}$

A proposed physically-based litter stock model was parameterized using a simple mass balance budget approach; e.g., Equation (6),

$$\text{MBM}_{LS} = LI - LO \pm \frac{\partial LS}{\partial t} \tag{6}$$

This model accounts for the change of inputs $\left(\int_{t=t0}^{t=tf} \frac{\partial LI}{\partial t}\, \partial t\right)$ from litter fall, the change of litter stocks $\left(\int_{t=t0}^{t=tf} \frac{\partial LS}{\partial t}\, \partial t\right)$ from all litter gains (litter fall), and all litter losses (decomposition, wildfires, surface runoff) and is solved to determine the change of outputs $\left(\int_{t=t0}^{t=tf} \frac{\partial LO}{\partial t}\, \partial t\right)$ from all litter losses (decomposition, wildfires, surface runoff), as summarized in Equation (7):

$$\left(\int_{t=t0}^{t=tf} \frac{\partial LO}{\partial t}\, \partial t\right) = \left(\int_{t=t0}^{t=tf} \frac{\partial LI}{\partial t}\, \partial t\right) \pm \left(\int_{t=t0}^{t=tf} \frac{\partial LS}{\partial t}\, \partial t\right) \tag{7}$$

The proposed model does not consider the incorporation of litter as organic matter to the soil as it has been estimated to be less than 2% of the litter stock mass balance and it is assumed this component remains within the litter stocks [2,41,42]. The integration of these equations from the time of planting to the time of measurements of each plantation results in $LI$-$LO \pm LS = 0$, where $LI$ = inputs from litter fall (Mg ha$^{-1}$ y$^{-1}$), $LO$ = outputs from litter losses (Mg ha$^{-1}$ y$^{-1}$), and $LS$ = litter stocks (Mg ha$^{-1}$ y$^{-1}$).

Litter fall, litter loss, and changes in litter stocks are a function of several site-based environmental and tree genetic variables including productivity, *AGB*, tree species, and diversity of tree species. The rate of litter fall (Mg ha$^{-1}$ y$^{-1}$) was reported to be a function of *AGB* (total mass of stem, stump, branches, and foliage) in the planted plots of northern Mexico [37]. The originally measured litter fall rate and *AGB* data (16 planted plots) were fitted to a linear regression equation (Litter fall = 1.24 + 0.14 *AGB*). In addition the sigmoid Chapman-Richards function shown in Equation (9) was fitted to account for 0 litter fall in 0-yr-old planted forests as well as to better physically depict the litter fall data as a function of *AGB* (Figure 2):

$$\frac{\partial LR}{\partial t} = 13.55(1 - \exp(-0.003AGB))^{0.48} \qquad (8)$$

where: $LR$ = litter fall rate (Mg ha$^{-1}$ y$^{-1}$).

In addition to the originally reported litter fall data, other data sets, including the confidence bounds, of the rate of litter fall [43–46]; other regional data, Mexico's commercial forest inventory data sets, and World litter data in forests [47] were added to the graph to test the strength of the relationship across these conifer forests.

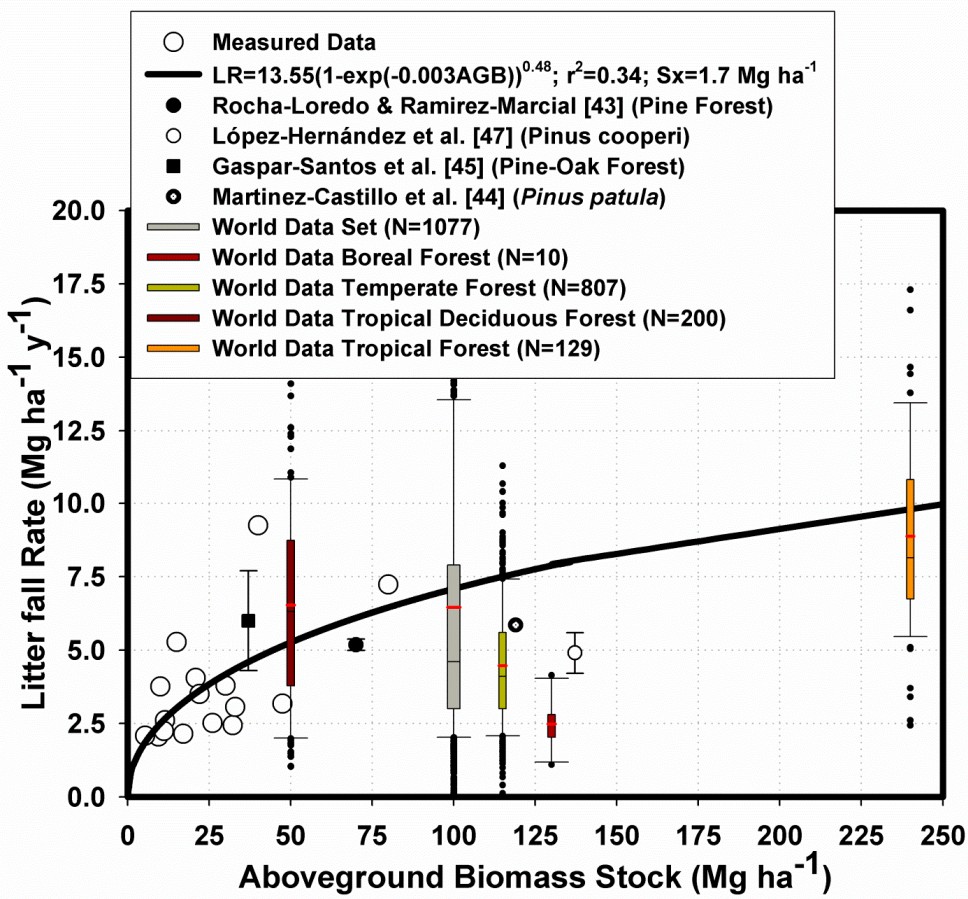

**Figure 2.** Predicted rates of litter fall as a function of Aboveground Biomass Stock (hollow dots) for northern Mexico's non-industrial plantations overlapped with some other local, regional, national and World data source (Original local data source: [37]).

Allometric equations previously reported [48,49] were applied at the individual tree level and summed the trees to calculate plot *AGB* (mass of timber, branches, and foliage) in each plot, then predicted plot *AGB* with timber stocks derived from the growth and yield model. Substituting the *AGB* Equation (9) into the Chapman-Richards Equation (8), and multiplying this equation times *AGB*, yields average litter fall ($\overline{LI}$) from time of planting to time of measurements as follows:

$$\overline{LI} = AGB \int_{AGBt = 0}^{AGBt = ti} 13.55 \left( 1 - \exp\left( -0.003 \cdot \beta_0 V^{\beta 1} \right) \right)^{0.48} \cdot \partial t \tag{9}$$

where *V* is mathematically derived in Equation (10):

$$Ln(V) = f(\text{BA}) \tag{10}$$

As an example, average litter fall rate, $\overline{LI}$, times dt yields cumulative litter fall $LF = \int_{AGBt = t0}^{AGBt = t1} \left( \overline{LI} \cdot \partial t \right)$; dt is the time interval from planting ($t_0$) to the time of measurements of litter stocks ($t_1$). By regressing litter stock losses (outputs = inputs-litter Stocks) versus litter fall (inputs) and weighting litter outputs by site index, the output of the $\text{MBM}_{\text{LS}}$ model is litter stocks, (*LS*), as in Equation (11):

$$LI - LO = \pm LS; \frac{LO}{LI}(LI) = f\left( LI, \frac{P}{Et}, \text{SI} \right); LI - (LO) = LS \tag{11}$$

This approach assumes in perturbed forest stands (e.g., by wildfires) litter stocks at the start time 0 were 0 and that environmental factors promote similar litter losses in all planted plots.

*2.5. Empirical Model Applications*

Litter rainfall interception was estimated as a function of simulated mass of litter stocks. Using reported data e.g., [50,51], litter rainfall interception (%) was regressed against the mass of litter stocks forcing the regression line to pass through the origin and the slope of this relationship predicts litter rainfall interception as a percentage of the annual precipitation. The fuel model for estimating fire behavior proposed earlier [24] was fitted to total measured litter stocks and the litter stocks projected with the best fitting model, as it has been tested for North American forests. This procedure allowed us to forecast the time required by forest plantations to shift to a risk factor for high-severity forest wildfires. The $\text{MBM}_{\text{LS}}$ model can be preliminarily used to derive litter losses and the portion of litter losses to decomposition and other sources.

**3. Results**

Summaries of the fundamental tree and litter stock data are reported in Table 1. The evaluated forest plantations were, on average, 16 years old and as a consequence they carried small amounts of litter stocks.

Field measurements recorded a mean (confidence interval) mass of litter stocks of 6.0 ($\pm$1.4) Mg ha$^{-1}$. After 16 years of being planted, the average tree density has already declined 36%, from a mean density (standard deviation) of 5500 ($\pm$4000) to an average of 3500 ($\pm$3500) trees per hectare. The fitted Weibull probabilistic function demonstrated that the distribution of litter stocks across all 133 plots was skewed, with maximum likelihood shape, scale, and location parameters of 0.38, 2.79, and 0.00027, respectively. Approximately 60% of all planted stands (80/133) recorded values of <12.0 Mg ha$^{-1}$. Only 2 out of 9 plots re-measured during 2014 had litter stocks > 29 Mg ha$^{-1}$, classified as the mature/over mature timber and understory fire regime, with a higher risk of high-severity wildfires.

**Table 1.** Statistics of measurements for plots used to fit and validate litter stock equations for northern Mexico forest plantations.

| | Ni | Nf | *t* | Db | H | HD | CC | BA | *V* | *AGB* | SI | Litter |
|---|---|---|---|---|---|---|---|---|---|---|---|---|
| | | | | | Plots (38) measured in 2002–2003 in Durango | | | | | | | |
| M | 4286 | 2108 | 18 | 14 | 7 | 6 | 9 | 30 | 73 | 50 | 6 | 12 |
| SD | 3735 | 1383 | 4 | 5 | 3 | 5 | 3 | 17 | 54 | 29 | 4 | 8 |
| CI | 1188 | 440 | 1 | 1 | 1 | 1 | 1 | 5 | 17 | 9 | 1 | 2 |
| | | | | | Plots (9) re-measured in 2014 in Durango | | | | | | | |
| M | 10,000 | 1234 | 36 | 17 | 12 | 14 | 10 | 45 | 147 | 71 | 10 | 21 |
| SD | 0 | 347 | 0 | 1 | 3 | 3 | 1 | 11 | 88 | 15 | 3 | 10 |
| CI | 0 | 67 | 0 | 0 | 1 | 1 | 0 | 2 | 17 | 3 | 1 | 2 |
| | | | | | Plots (46) measured in 2002–2003 in Nuevo Leon | | | | | | | |
| M | 10,000 | 7534 | 6 | 2 | 1 | 0 | 1 | 3 | 5 | 3 | 6 | 1 |
| SD | 0 | 2526 | 6 | 3 | 2 | 1 | 2 | 5 | 11 | 8 | 5 | 4 |
| CI | 0 | 730 | 2 | 1 | 1 | 0 | 1 | 2 | 3 | 2 | 1 | 1 |
| | | | | | Plots (40) measured in 2002–2003 in Coahuila | | | | | | | |
| M | 1939 | 418 | 26 | 16 | 4 | 3 | 5 | 11 | 17 | 14 | 10 | 4 |
| SD | 1523 | 192 | 11 | 9 | 2 | 1 | 3 | 8 | 16 | 12 | 3 | 3 |
| CI | 504 | 64 | 4 | 3 | 1 | 0 | 1 | 3 | 5 | 4 | 1 | 1 |

where: Ni = initial planting density ($n$ ha$^{-1}$); Nf = final stand density at the time of measurements ($n$ ha$^{-1}$); $t$ = age (years); Db = mean basal diameter (cm); H = mean top height (m); HD = mean top height of dominant trees (m); CC = mean canopy cover of individual trees (m$^2$); BA = basal area (m$^2$ ha$^{-1}$) measured with basal diameter; $V$ = stand timber volume (m$^3$ ha$^{-1}$); $AGB$ = Aboveground biomass (Mg ha$^{-1}$); Litter = Litter stocks (Mg ha$^{-1}$); M = mean, SD = standard deviation, CI = confidence intervals ($\alpha = 0.05$, $v = n - 1$).

### 3.1. Plot-Scale Variables and Models

Most plot-scale variables (mean basal diameter, basal area, density, site index, and timber volume) depicted two distinctive clusters of data. The forest plantations in Durango presented higher values of productivity, basal diameter, basal area, timber volume and aboveground biomass. For the simplification of prediction of litter stocks in these planted forests, common statistics and models for all plots were derived, for all species, for the Sate of Durango. Forest plantations of Coahuila and Nuevo Leon do not accumulate large amounts of litter and therefore they do not intercept significant volumes of precipitation neither they sustain major high-severity forest wildfires. However, litter stock data for Nuevo Leon and Coahuila was plotted together with measured and predicted data by the two models.

Dominant trees reached an average height of 8 m at 15 years of age (Figure 3).

There is an ample spread within the data for the top height of dominant trees with SI15 extremes of 2.6 and 10.6 m, showing the stand productivity determined by the site index approach classified these planted forests well. Goodness of fit statistics showed that a little over 70% of the total variance was explained by the sigmoid Chapman–Richard site index equation. Large mortality rates of seedlings ($5491 - 3529/5491 = 35\%$) after 16 years of planting, by a variety of stochastic (light wildfires, pests and diseases, trampling) and deterministic (competition) factors, resulted in differential mortality rates and depicted a large spatial variability of tree densities. Stand density plays a key role in the height dominant trees reach at base age. Due to differential causes of mortality, with the same age some plots had overstocked while other plots showed under-stocked stand density modifying drastically top height of dominant trees.

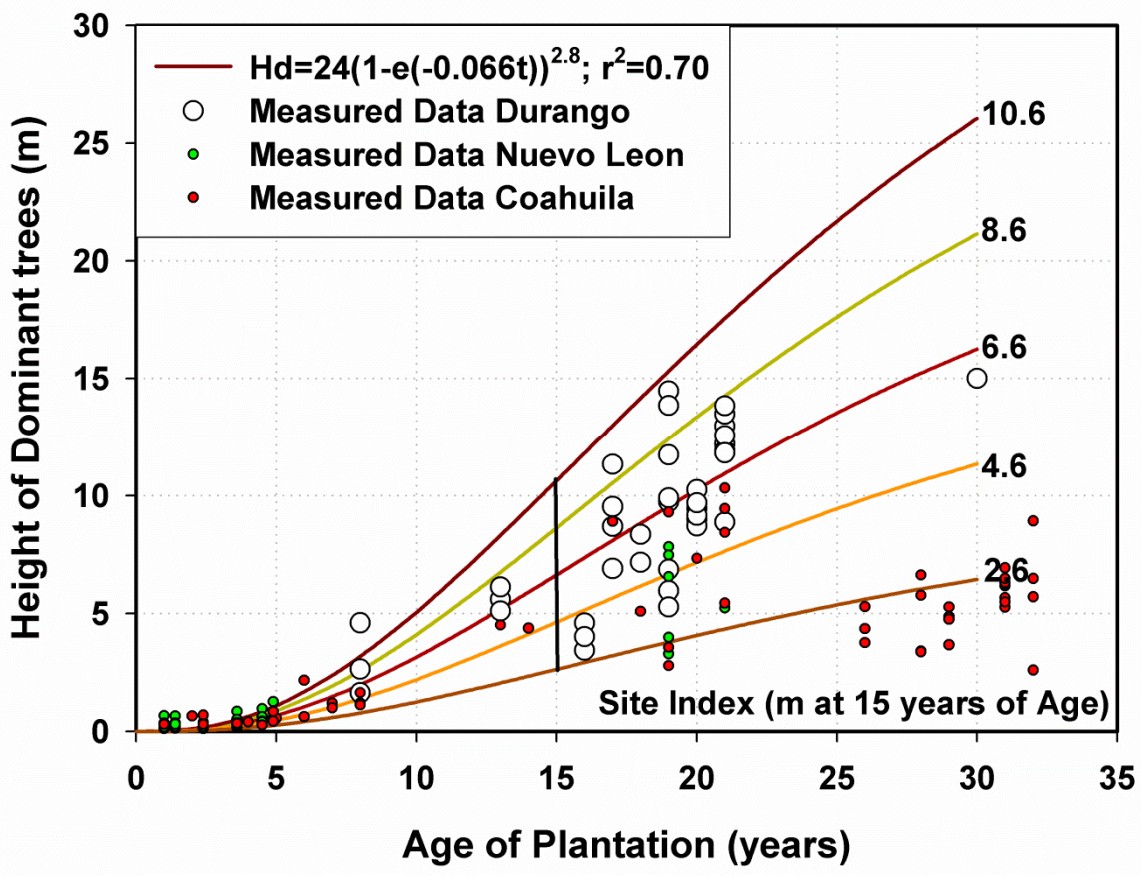

**Figure 3.** Evaluation of site index using the Chapman-Richards equation for northern Mexico's non-industrial planted forests with site index values inside for each curve. Lines represent the site index curves.

Basal Area Growth Model

The basal area growth model of Equation (1) uses only the plot-scale variable plantation age (Table 2). Because basal area was forced to reach 50 m$^2$ ha$^{-1}$ in 80-year-old plantations, the growth model did not explain a significant portion ($r^2 = 0.24$; $p = 0.0025$) of the large basal area deviance for any individual of the three data sources (Table 2).

**Table 2.** Plot-level parameters and the volume growth and yield model developed to litter stocks and accumulation rates of Northern Mexico non-industrial forest plantations.

| Equation (No.) | Parameter | SM$_{LS}$ System of Equations | $r^2$ | Sx |
|---|---|---|---|---|
| (1) | Basal Area (m$^2$ ha$^{-1}$) | $BA = \left[86(5.8)(1 - e(-0.043(0.01)t))^{1.64(0.21)}\right]$ | 0.24 | 13 |
| (2) | Timber Volume (m$^3$ ha$^{-1}$) | $V = \exp(1.0697 + 1.5329Ln(BA))$ | 0.52 | 237 |
| (3) | AGB (Mg ha$^{-1}$) | $AGB = \exp(1.4122 + 0.5929Ln(V))$ | 0.90 | 424 |
| (4),(5) | Litter Stocks (Mg ha$^{-1}$) | $SM_{LS} = \exp\left(-1.7819 + 3.1356Ln\left[\frac{P}{Et}\right] + 1.1494Ln(AGB)\right)$ | 0.71 | 4.2 |
| | | **MBM$_{LS}$ System of Equations** | | |
| (6) | Litter Fall Rate (Mg ha$^{-1}$ y$^{-1}$) | $LI = 13.55(1 - \exp(-0.003AGB))^{0.48}$ | 0.34 | 1.7 |
| (7) | Litter Outputs (Mg ha$^{-1}$) | $LO = \exp\left[-0.85 + 1.12 \times Ln(LI) - 0.77 \times Ln\left[\frac{P}{Et}\right]\right]$ | 0.97 | 3.1 |
| (8) | Litter Stocks (Mg ha$^{-1}$) | $MBM_{LS} = \left[\int_{AGBt\,=\,0}^{AGBt\,=\,t1} LI \cdot \partial t\right] - \exp\left[-0.85 + 1.12\,Ln(LI) - 0.77 \times Ln\left[\frac{P}{Et}\right]\right]$ $MBM_{LS} = \int_{t\,=\,0}^{t\,=\,t} LI \cdot \partial t - \int_{t\,=\,0}^{t\,=\,t} LO \cdot \partial t$ | 0.32 | 6.0 |

BA = Basal area (m$^2$ ha$^{-1}$); $V$ = timber volume (m$^3$ ha$^{-1}$); $t$ = time or age of forest plantations (years); $AGB$ = tree aboveground biomass (Mg ha$^{-1}$); $LI$, $LO$ = Litter inputs and outputs, respectively (Mg ha$^{-1}$); $P$ = Average annual precipitation (mm); $Et$ = average annual Thornthwite evapotranspiration (mm); $r^2$ = Coefficient of Determination; Sx = Standard Error.

The model predicts that, for average productivity stands at 40 years of age, forest plantations would bear basal areas of 40 m$^2$ ha$^{-1}$, a statistic slightly smaller than that found the in re-measured Durango plots from which the 2014 field campaign reported an average (standard deviation) of 45 ($\pm$8 m$^2$ ha$^{-1}$) in 36-yr-old plantations.

### 3.2. SM$_{LS}$ Empirical Model

The stepwise procedure selected the ratio of $P/Et$ over the SI15 as statistically significant variable ($p = 0.0002$) describing the mass of litter stocks, with the regression equation and parameters described in Table 2. The conventional site index variable developed by Equation (2) did not explain a significant portion of the SM$_{LS}$ variability ($p = 0.38$). Plot timber volume, $V$, was evaluated using $BA$ resulting in the equation reported in Table 2. The $BA$, $P/Et$, and $V$ equations depicted in Table 2 are part of the growth and yield model. The productivity index is employed once the SM$_{LS}$ model parameters were developed. The SM$_{LS}$ model in logarithmic transformation fits the litter stock data well, with $r^2$ values of 0.84 and 0.79 for fitting and validating datasets, respectively (Figure 4). The SM$_{LS}$ equation with $AGB$ values > 55 Mg ha$^{-1}$ in high productivity stands ($P/Et > 1.2$) records litter stocks >29 Mg ha$^{-1}$, and were, thus, classified as having higher risk of large-scale wildfires. 20-year-old planted plots in high productive stands ($P/Et > 1.2$) would have litter stocks $\geq$ 29 Mg ha$^{-1}$, with BA > 23 m$^2$ ha$^{-1}$ (Figure 4).

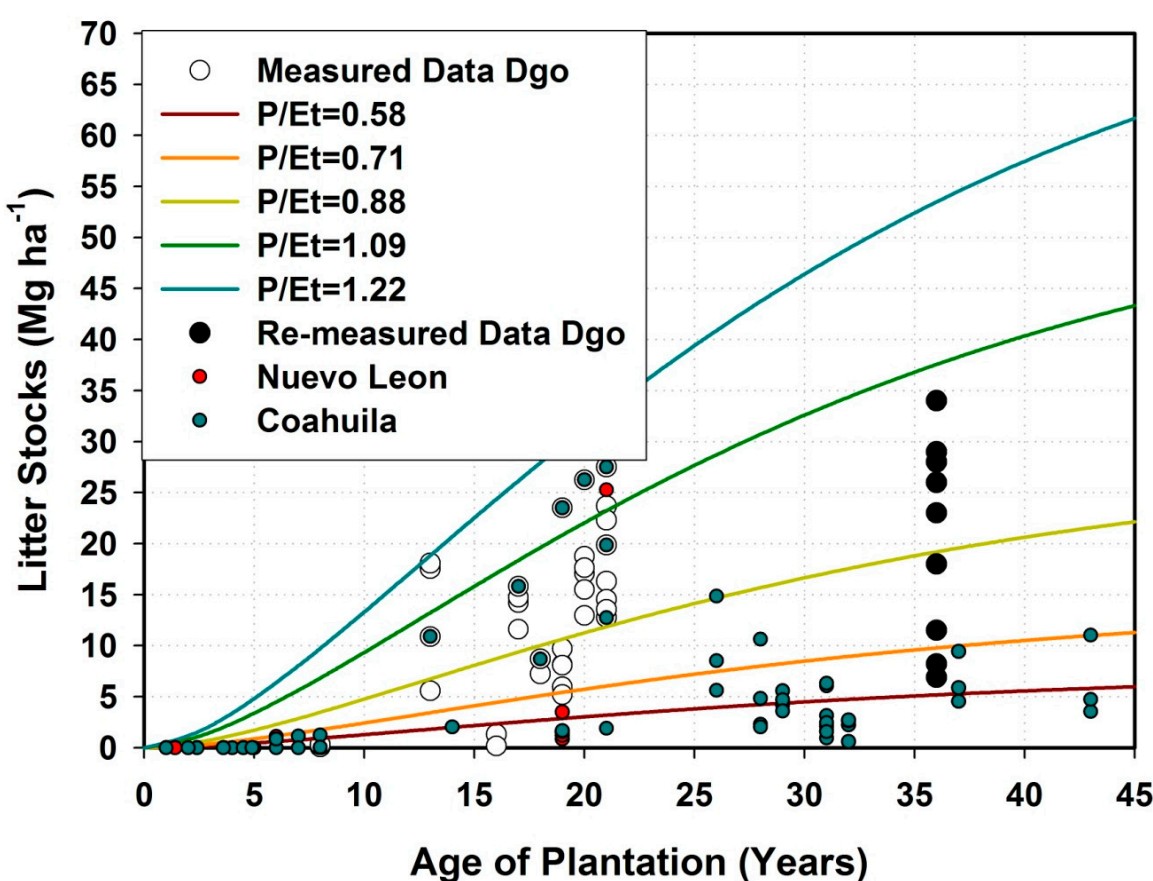

**Figure 4.** The statistical, SM$_{LS}$, model for evaluating litter stocks and accumulation rates for non-industrial planted forests of northern Mexico.

### 3.3. MBM$_{LS}$ Model

The sigmoid Chapman–Richards function (Litter fall = 13.55(1 − exp(−0.003$AGB$))$^{0.48}$) evaluated the maximum annual rate of litter fall as a function of total live standing $AGB$, which would be less than 8.0 Mg ha$^{-1}$ y$^{-1}$ (Figure 2). This value falls within the data.

Equation (10) predicts *AGB* values of 55 and 93 Mg ha$^{-1}$ in forest plantations with basal areas of 23 m$^2$ ha$^{-1}$ and 41 m$^2$ ha$^{-1}$, respectively.

In these plantations, cumulative litter fall represents a significant input of organic matter to the topsoil and cumulative litter losses represent a significant output of litter stocks. 20 and 40-year-old forest plantations with average site productivity ($P/Et = 0.80$) should accumulate 75 and 200 Mg ha$^{-1}$ of litter fall, with an annual averages of input rates of 3.75 and 5.00 Mg ha$^{-1}$ y$^{-1}$, respectively However, the field measurements of the mass of litter stocks in 20-year and projection of 40-year-old plantations with average site productivity was only 18 Mg ha$^{-1}$ and 39 Mg ha$^{-1}$, respectively. Total accumulated litter outputs in 20 and 40-year-old plantations reach 57 and 161 Mg ha$^{-1}$; with average annual flux losses of 2.85 and 4.03 Mg ha$^{-1}$ y$^{-1}$, respectively of the annual average rate of litter fall, only an average of 0.95 Mg ha$^{-1}$ y$^{-1}$ of litter fall accumulated every year as litter stocks. That is, the litter fall and litter losses control the mass balance model and they are closely related fitting Equation (11) (Table 2).

Equation (11) forecasts that in 40-year-old forest plantations approximately only 19% of litter fall inputs remain on the soil as litter stocks. The last equation of Table 2 represents a simplification of the MBM$_{LS}$ model. Both models display correctly the measured and re-measured litter stock variability for the forest plantations of Durango, Nuevo Leon, and Coahuila (Figures 4 and 5). It projects 23-year-old plantations would have litter stocks >29 Mg ha$^{-1}$ in high productivity stands ($P/ET > 1.2$). Forest plantations growing on poorly-productivity stands ($P/Et \leq 0.90$) would never accumulate significant litter stocks.

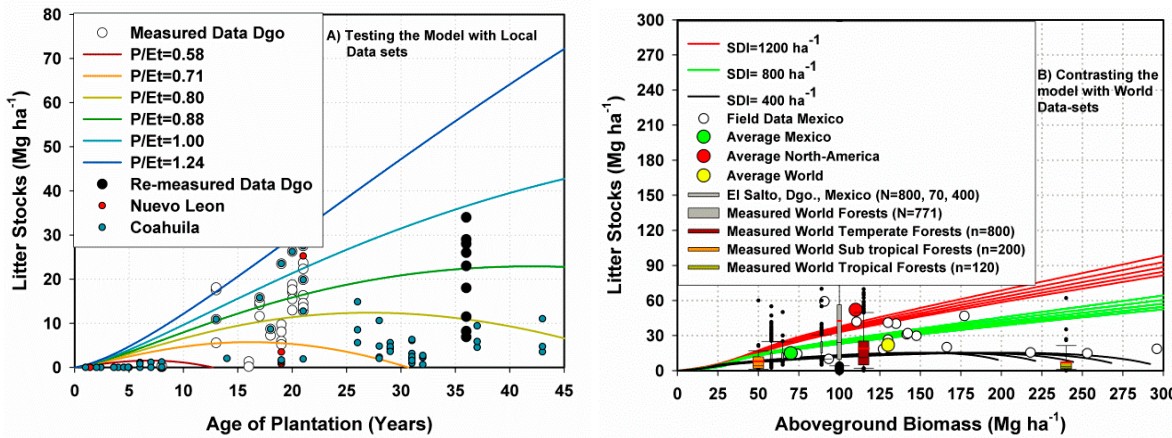

**Figure 5.** The mass balance budget model, MBM$_{LS}$, for evaluating litter stocks and accumulation rates for (**A**) non-industrial planted forests of northern Mexico overlapped with (**B**) some other regional, national and World litter stock data source. SDI = stand density index with five different site index curves for each SDI.

### 3.4. Comparisons of Model Performance

The statistics and models for all plots, for all species, for the State of Durango are reported in Table 3.

**Table 3.** Goodness of fit statistics for two models of predicting litter stock accumulation data in non-industrial northern Mexico's reforestations.

| Model | | $n$ | $r^2$ | Sx (Mg ha$^{-1}$) |
|---|---|---|---|---|
| Statistical Model, SM$_{LS}$<br>$SM_{LS} = \exp\left(-1.7819 + 3.1356 Ln\left[\frac{P}{Et}\right] + 1.1494 Ln(AGB)\right)$ | Fit ($n = 38$) | 47 | 0.84 | 5.1 |
| | Validation ($n = 9$) | 9 | 0.79 | 8.3 |
| Mass Balance Budget Model, MBM$_{LS}$<br>$MBM_{LS} = \int_{AGBt=0}^{ABGt=t} LI \cdot \partial t - \int_{t=0}^{t=t} LO \cdot \partial t$ | Fit ($n = 38$) | 47 | 0.32 | 6.5 |
| | Validation ($n = 9$) | 9 | 0.66 | 6.1 |

where: $r^2$ = coefficient of determination; Sx = Standard Error.

In spite of the large *LS* variability, litter stocks were significantly predicted by both models ($P$ = 0.0001) but they were better predicted by the $SM_{LS}$ than by the $MBM_{LS}$ model. The variance explained by the models, $r^2$, increased from a mean of 0.32 to 0.84 and the standard error, Sx, was reduced from a mean of 6 Mg ha$^{-1}$ to 4.2 Mg ha$^{-1}$ from the $MBM_{LS}$ to the $SM_{LS}$. Regardless of the unexplained variation of these two kinds of models, the predicted *LS* data by the $MBM_{LS}$ model covered well the measured and re-measured litter stock data for the Durango; as well as for the Coahuila and Nuevo Leon plantations. Predicted litter stock data with the $MBM_{LS}$ model tended to curve more in poorly productive stands, although the variance explained by the model increased from 32% when fitting to 66% when validating the model. This model also predicts litter stocks would decay to 0 values over time in poorly productive forest stands. A mass of close to 0 litter stocks was observed in poorly productive planted forests of Durango even though they were 16-yr old. Several forest plantations of Coahuila and Nuevo Leon also recorded very small *LS* values because the ratio $P/Et$ < 1.0. Hence, the $MBM_{LS}$ model is recommended for further projecting litter stocks and accumulation rates in forests prior to evaluate better stand productivity and litter fall of forests. Stand productivity is better modeled using the ratio of $P/Et$ than by the conventional site index equation but any other more precise water balance budget model would improve predictions.

### 3.5. Model Applications

Litter rainfall interception can be predicted by multiplying the predicted mass of litter stocks by a factor of 0.45. That is, each tonnage of litter stocks intercepts approximately 0.45% of the annual rainfall. Northern temperate forests of Durango, Mexico have recorded an annual average of 1000 mm of precipitation indicating that 45-year old forest plantations in high-productivity stands ($P/Et$ > 1.0) would bear 40 Mg ha$^{-1}$ of litter stocks that would intercept 18% (180 mm) of the average annual precipitation. Consistent with measurements in the second field inventory, the $SM_{LS}$ predicted that 38-year-old forests growing within temperate forests in medium to high productivity stands would accumulate litter stocks of more than 29 Mg ha$^{-1}$ in high productive stands ($P/Et$ > 1.0) posing a high-risk threat of large-scale wildfires.

## 4. Discussion

### 4.1. Contrasting the Model Performance

The $SM_{LS}$ model predicted better litter stocks of planted forests, although the $MBM_{LS}$ is physically-based and it can predict better litter stocks in other forests. Both models display the full variance of the *LS* dataset for forest plantations of Durango, Nuevo Leon, and Coahuila, Mexico. The later model projects 0 litter stocks in poorly productive forest stands, and computes litter inputs, outputs and changes in litter stocks. Physically, tree *AGB* is made of the mass of timber, branches and foliage [4]. These biomass components are the main sources of litter fall in the form of stems, branches, bark, and foliage; which in turn are the main inputs of litter stocks. The associations between tree biomass components, litter fall and litter stocks clarify the good correlation between tree *AGB* and the annual rate of litter fall depicted by the sigmoid Chapman-Richards function as well as between tree *AGB* and LS ($LS$ = 15.15 (1 − exp (−0.162 × $AGB$))$^{8.6}$; $r^2$ = 0.50). The $MBM_{LS}$ model derives litter outputs based on the statistical relationship between litter fall inputs and litter stocks, in the absence of measurements of litter losses. This model provides good predictions of litter stocks, losses and inputs for the planted forests of Durango and reproduces well the *LS* for the forest plantations of Nuevo Leon and Coahuila, as well. The $MBM_{LS}$ model has also the advantage over the $SM_{LS}$ model that it also provides insights into the fate of litter losses from decomposition and the residual losses by forest wildfires and surface runoff events. The $MBM_{LS}$ model could be improved if the annual litter fall rate predicted by the sigmoid Chapman-Richards model is weighted by other productivity indices, or by deriving an individual basal area growth model for each of the two clusters (Durango and Coahuila-Nuevo Leon) observed in the plot data using data from older

forest plantations. The $SM_{LS}$ model, on the other side evaluates well litter stocks for the plantations of Durango but performs poorly when projecting 0 litter stocks commonly found for several of the plantations of Coahuila and Nuevo Leon. It is likely this statistical model would also projects biased litter stock assessments for other planted forests in degraded, low-productivity sites.

At this time the $MBM_{LS}$ model is semi-empirical in nature and matches well measured and evaluated litter stocks of northern forest plantations of Mexico. The model can be easily refined or expanded to predict the major litter losses such as decomposition, wildfires and surface runoff. Meanwhile the $MBM_{LS}$ model requires local calibration of: (a) the *AGB* growth and yield model; (b) the relationship between the rate of litter fall and tree *AGB*; and (c) the relationship between cumulative litter fall and litter stocks to derive litter losses in case litter outputs are not being measured on site or derived from other more physical relationships. The use of the $MBM_{LS}$ model without local calibration could bias the evaluation of litter rainfall interception and fire risk in high productive stands.

### 4.2. Litter Management

The $MBM_{LS}$ model projects 25-year-old forest plantations in high productivity stands ($P/Et > 1.0$) would yield 40 Mg ha$^{-1}$ of litter stocks that would intercept approximately 18% of the annual precipitation in northern temperate forests of Durango, Mexico (180 mm). This depth of precipitation can be reduced by carrying out management practices to reduce litter stocks to e.g., 10 Mg ha$^{-1}$ and save 135 mm of precipitation. However, at this time, it is difficult to quantify the effect of this practice on soil water evaporation that should be kept to a minimum as a goal of saving 135 mm of annual precipitation and improve the water balance of forests. Data on soil water evaporation under different litter stocks masses is required to complete this balance and to predict the mass of litter stocks where the water balance can reach its maximum efficiency. This is a matter of further research. For northern coniferous forests of Mexico, Alanís et al. [18] recommended reducing litter stocks using prescribed burning approaches would not increase surface runoff neither soil erosion rates when leaving a litter bed of at least 2 cm of depth that is equivalent to approximately 15 Mg ha$^{-1}$ of LS.

Litter management practices including prescribed burning, litter collection, litter arrangement, and redistribution are becoming common practices in several forests. $MBM_{LS}$ predicts forest plantations accumulate litter stocks >29 Mg ha$^{-1}$ when they reach timber volume or biomass of 125 m$^3$ ha$^{-1}$ or 70 Mg ha$^{-1}$, when they are 23 years old in high productive stands ($P/Et > 1.2$). Litter stock mass > 29 Mg ha$^{-1}$ was already measured in planted plots during 2014. At the present, productive native coniferous forests often carry litter stocks >29 Mg ha$^{-1}$ but other sources of litter stock variability have also been measured and reported [18,22]. The $MBM_{LS}$ models for Durango's forest plantations predict similar mean litter stocks of between 40 and 50 Mg ha$^{-1}$ with similar basal areas. The model predicts litter stocks of 100 Mg ha$^{-1}$ in high-productive forests with 350 m$^3$ ha$^{-1}$ of timber stocks consistent with measurements of large masses of litter stocks in other world forests [29–31].

### 4.3. The MBM_LS Mass Balance Budget Model

Decomposition is also a major process responsible for the high litter losses in these planted forests. It is regulated by several variables including the litter's physical and chemical properties, climate, nutrient composition, forest activities, and macro- and micro faunal responses [26,27,52–54]. Mean decay rates have been estimated to vary between 0.07% to 0.16% ($\pm$0.002%) per day using leaf-litter bags [25,55]. The former value represents the daily decomposition rate for Arizona pine forests. Using this value, the cumulative decomposition rate of the litter fall would have a total figure of 50 Mg ha$^{-1}$ y$^{-1}$ in 40 year old forest plantations. The $MBM_{LS}$ model projects the 40 year old planted forests to have an estimated mean cumulative litter fall of 193 Mg ha$^{-1}$ (150 Mg ha$^{-1}$ foliage and 43 Mg ha$^{-1}$ branches) and average litter stocks of 37 Mg ha$^{-1}$, with total cumulative litter losses of

156 Mg ha$^{-1}$ (121 Mg ha$^{-1}$ foliage and 35 Mg ha$^{-1}$ branches). Neither the SM$_{LS}$ nor the MBM$_{LS}$ model predicts litter decomposition but a weighted factor of 40% of all evaluated litter losses by the MBM$_{LS}$ model could be a good decomposition leaf weighted factor (50 Mg ha$^{-1}$/121 Mg ha$^{-1}$) for these models. Decomposition rates for the northern forests of Mexico may differ from those measured elsewhere, such as northern Arizona´s forests, and other causes such as fire and surface runoff may have affected the litter stock balance.

Litter decomposition likely differs between the three major plantation sites as it is partially regulated by climate variables. Annual potential evapotranspiration (*Et*) and precipitation (*P*) vary between these three major plantation sites: mean values for Durango are 600 mm and 1000 mm (*P*/*Et* = 1.67); 800 mm and 600 mm (*P*/*Et* = 0.75) for Nuevo Leon; and 900 mm and 500 (*P*/*Et* = 0.55) for Coahuila. The arid and semi-arid ranges of Coahuila and Nuevo Leon with mean annual temperatures in the range of the 18–22 °C showed the largest mean *LS* loss factor (93%), in contrast to the smaller *LS* loss factor of temperate forests of Durango (81%), which presents mean annual temperatures in the range of 12–14 °C.

Litter decomposition variability can also be accounted for by other factors. For example, litter quality is a major control in several forests [56]. Nitrogen and lignin content, the Klason ratio, and litter diversity all affect litter quality and play an important role in decomposition rates [26,27,52,57]. Different planted pine species have different values for lignin and nitrogen content and the Klason ratio, all of which may also contribute to the total litter stock decomposition variance. The likely unexplained litter decomposition variance of the MBM$_{LS}$ must also be related to litter quality; breaking it into several compartments may help to better predict this process. Litter fall in the forests of northern Mexico includes woody material (~23%) which is not a substrate used in the litter decomposition models.

Forest wildfires and surface runoff events may explain the remaining cumulative 71 Mg ha$^{-1}$ of litter loss composed of leaf and woody debris (43 Mg ha$^{-1}$) and 35 Mg ha$^{-1}$ of branches and necromass in 40-year-old forest plantations that is un-accounted for by decomposition. When measuring litter stocks, several standing trees and branches remaining on soils showed signs of past small-scale forest wildfires. The mass of litter burned is hard to quantify as it is dependent on several litter features, plus its moisture content, fire intensity and rate of spread [18,58,59].

Surface runoff is an infrequent and spatially-limited event while saturated throughflow takes place more frequently in the temperate forests of northwestern Mexico [18,60,61]. However, during the rainy season, rivers drain upland mountain temperate forest watersheds with high litter loads and their source may vary between sites and locations. The amount of litter running off depends on several factors such as ground vegetation coverage, needle versus broadleaf litter, among others that require further study to better parameterize the full litter stock balance [58]. For example, Abelho, [62] reported forests lose litter from 5.0 to 9.2 Mg ha$^{-1}$ y$^{-1}$ to runoff.

Therefore, there is an urgent need for measuring litter fall, litter loss, and stock changes more precisely on permanent sampling plots in order to collect field data and to solve the MBM$_{LS}$ model with a higher precision than the approach followed in this research. Timber harvesting produces large fresh litter and branch stock inputs onto forest soils. This practice is often carried out in most of the northern coniferous forests of Mexico and the amount of litter stocks added to the ground can be predicted in advance. However, stochastic events that control large litter fall and necromass inputs, such as high-severity droughts, heavy frosts, strong winds, and cold front systems associated with snow, must also receive full attention in future litter stock evaluations and assessments, as they may carry a major risk factor for starting high-severity, widespread forest wildfires. However, a great deal of litter fall uncertainty is expected within each as well as among these stochastic events.

## 5. Conclusions

The measurement and modeling of litter stock and litter accumulation rates is useful for setting appropriate management practices to meet several objectives, including

improving the water balance of forests and reducing the risk of high-severity forest wildfires. In three study regions of northern Mexico, litter stock measurements recorded a mean of 6.0 ($\pm$1.4) Mg ha$^{-1}$ for 16 year old plantations on average-productivity stands; resembling the closed, short-needle timber litter model where slow-burning ground fires with low flame lengths is a common fire regime. However, two developed models predict, consistent with field observations that 23 year old planted forests in high productivity stands intercept 15% of annual rainfall and shift to a new fire regime classified as high risk of high-severity wildfires because predicted litter stock mass >29 Mg ha$^{-1}$. Therefore, litter stock management practices such as prescribed burning, litter collection, and litter arrangement are some recommended practices at this stage of tree growth or in native forest stands with timber volume or aboveground biomass stocks greater than 125 m$^3$ ha$^{-1}$ or 70 Mg ha$^{-1}$, respectively.

**Author Contributions:** F.d.J.R.-F. data curation, funding acquisition. J.-G.C. funding acquisition. J.d.J.G.-L. funding acquisition. J.N. conceptualization, formal analysis, writing-original draft, project administration. All authors have read and agreed to the published version of the manuscript.

**Funding:** This research received no external funding with the exception of parts of the publication costs (UNIPOLI and COCyTED).

**Data Availability Statement:** Data is available upon request at José Návar.

**Conflicts of Interest:** The authors declare no conflict of interest.

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
