# Peer review of "Modeling Litter Stocks in Planted Forests of Northern Mexico"

_forests, doi:10.3390/f13071049_

Round 1
Reviewer 1 Report
In my opinion, this research analyzed the decomposition of litter in different forests in northern Mexico through mathematical model, which is of great significance. I think it is well written, has appropriate icons, and reliable conclusion.
Author Response
Dear Reviewer I;
I greatly appreciate your constructive evaluation of this State of the Art on Modelling Litter stocks and accumulation rates in forests that can be further expanded to other world forests. Seems like your evaluation report positively recommends its publication as all major evaluation points are positively, 'YES', marked.
Kind Regards
Sincerely
Jose Navar, PhD
Reviewer 2 Report
General comments:
The manuscript “Modeling Litter Stocks and Accumulation Rates in Planted Forests of Northern Mexico” by Rodríguez-Flores et al. presents another approach to address litter stocks and accumulation rates in Northern Mexico forests. The study materials are valuable to guide a better forest management since those factors are the key factor of wildfire as fuels (or source of fire) and its indirect effect on water cycling in forest ecosystems.
However, the manuscript has to be a better framed. Honestly, it is difficult to follow the idea that the authors convey.
For example, the abstract needs to have 1) more explanation about background, 2) concise explanations of results and 3) implication or conclusion. However, this version of manuscript is not sufficient to address their purpose. In line 31-33, it is difficult to get the authors’ point. SMLS did a better job to predict, but MBMLS revealed relationships between litter decomposition and total litter loss. If SMLS did a better job, how can MBMLS reveal those relationships? Also, in the end of abstract, the authors insisted that litter managements need to be well managed to improve water balances and reduce the risk of wildfires. But there is no explanation of water balance and wildfire in the abstract.
These points in the structure of abstract are very similar to the rest of manuscript. Some important topics were not well addressed, but pop up without well explanations. While the research subject and approach would be beneficial for forest managements, overall structures of manuscript and writing must be substantially improved.
Specific comments:
Materials and methods: there are some sentences starting with ‘I’. This should be ‘we”.
Fig. 2: the label of “Pinus cooperi” contained unreadable letter
Results:
L 319-329: It is very difficult to get the explanation. Is it possible to create some plots to address those points? Table 1 contains lots of information with abbreviation, but it is difficult to catch the authors’ points.
L 349: Is there any criteria why the authors mentioned mean height of 8 m at 15 years of age?
Table 2: What do you think about very lower R2 of basa area estimations? This lower R2 would be propagated to further applied equations
Fig. 5: need to have detailed legend, e.g., labeling panels (like a and b) and explain each panel.
Author Response
DEAR REVIEWER 2;
I GREATLY APPRECIATE YOUR HELP AND CONSTRUCTIVE CRITICISMS ON THE MANUSCRIPT TO IMPROVE ITS TECHNICAL CONTENT. RESPONSE TO COMMENTS ARE EMBEDDED IN CAPITAL LETTERS BELOW.
The manuscript “Modeling Litter Stocks and Accumulation Rates in Planted Forests of Northern Mexico” by Rodríguez-Flores et al. presents another approach to address litter stocks and accumulation rates in Northern Mexico forests. The study materials are valuable to guide a better forest management since those factors are the key factor of wildfire as fuels (or source of fire) and its indirect effect on water cycling in forest ecosystems. THANKS FOR THIS ENCOURAGING COMMENT. JUST TO PRECISE: MODELING LITTER STOCKS IN THIS REPORT APPEAR TO BE PIONEER IN THE SCIENTIFIC LITERATURE.
However, the manuscript has to be a better framed. Honestly, it is difficult to follow the idea that the authors convey. I WILL DO MY BEST TO FRAME IT BETTER.
For example, the abstract needs to have 1) more explanation about background, BETTER BACKGROUND HAS BEEN ADDED 2) concise explanations of results RESULTS IS ABOUT TESTING THE MODELS AND RECOMMEND ONE AS IT IS DONE IN THE MANUSCRIPT and 3) implication or conclusion A BETTER CONCLUSION OR IMPLICATION HAS BEEN ADDED IN THE CONCLUSION SECTION. However, this version of manuscript is not sufficient to address their purpose. In line 31-33, it is difficult to get the authors’ point. SMLS did a better job to predict, but MBMLS revealed relationships between litter decomposition and total litter loss. YES SM DID A BETTER JOB WHEN PREDICTING AND VALIDATING THE MODELS BUT IT IS A STATISTICAL. AS WE UNDERSTAND BETTER BIOLOGY AND MATCH IT WITH MATH AS A PRELIMINARILY APPROACH BY THE MBM WE WILL COME UP WITH A BETTER TECHNOLOGY. If SMLS did a better job, how can MBMLS reveal those relationships? Also, in the end of abstract, the authors insisted that litter managements need to be well managed to improve water balances and reduce the risk of wildfires. But there is no explanation of water balance and wildfire in the abstract. THERE IS NOW EXPLANATION OF THE WATER BALANCE AND WILDFIRE IN THE ABSTRACT AS WELL AS IN THE INTRODUCTION AND RESULTS.
These points in the structure of abstract are very similar to the rest of manuscript. Some important topics were not well addressed, but pop up without well explanations. I MODIFIED THE MANUSCRIPT TO FOLLOW YOUR FRAME. HOPE I DID A GOOD JOB. While the research subject and approach would be beneficial for forest managements, overall structures of manuscript and writing must be substantially improved. THE MANUSCRIP HAS BEEN SENT FOR REVIEW TO SEVERAL COMPANIES WITH A POOR JOB IN IMPROVING THE ENGLISH AS WE NEED A FORESTER OR AN ECOLOGIST WHOSE NATIVE LANGUAGE IS ENGLISH.
Specific comments:
Materials and methods: there are some sentences starting with ‘I’. This should be ‘we”. DONE. THANKS.
Fig. 2: the label of “Pinus cooperi” contained unreadable letter. FIGURES HAD CHANGED FORMAT FROM SIGMA TO TIFF TO SHOW BETTER WHAT IS IN THE FIGURE.
Results:
L 319-329: It is very difficult to get the explanation. Is it possible to create some plots to address those points? Table 1 contains lots of information with abbreviation, but it is difficult to catch the authors’ points. NOW: N IS FOR DENSITY OR NUMBER, i AND f FOR INITIAL AND FINAL; D FOR DIAMETER AND H FOR TOP HEIGHT. THIS IS THE CODE USED BY FORESTERS
L 349: Is there any criteria why the authors mentioned mean height of 8 m at 15 years of age? SI SHOULD BE WITHIN THE DATA AND WE CHOSE THE AVERAGE AGE.
Table 2: What do you think about very lower R2 of basa area estimations? This lower R2 would be propagated to further applied equations. NO PROBLEM AS: (a) TO MODEL WAS FORCED TO PASS TRHOUGH THE AVERAGE BA; (b) TO BE CONSISTENT WITH THE FORMER BA MODEL FOR NATIVE FORESTS AND (c) FINAL LITTER STOCK EVALUATIONS WHEN CONTRASTED WITH OTHER REGIONAL, MEXICAN, AND WORLD DATA SHOWED EXCELLENT FIT.
Fig. 5: need to have detailed legend, e.g., labeling panels (like a and b) and explain each panel. I DID EXPLAINED BETTER PANEL A AND PANEL B IN THE CAPTION. THANKS
Reviewer 3 Report
Forests Review
Modeling Litter Stocks and Accumulation Rates in Planted Forests of Northern Mexico
Rodriguez-Flores et al.
Summary
This paper uses a statistical and mass balance model to predict litter stocks in planted forests in northern Mexico. Litter on the forest floor plays an extremely important role in the forest in terms of carbon cycling and maintaining moisture, however is also linked to wildfire hazard. The aim of this research is to develop models to assess litter stocks, accumulation and loss rates using statistical and mass balance approaches. Their results showed that the statistical model provided better predictions. Their results also show that high productivity 40 year old stands accumulate sufficient litter that can shift the forest to a new fire regime with a risk for high severity forest wildfires. This research is of significance to the field, capturing a process that has not been sufficiently studied and has great implications for forest ecology. However, there are many instances of poor organization and grammar that the manuscript, and thus, in its current form is not ready for publication.
General comment:
I really like the use of the statistical and mass balance model approach in this paper, and I think there is a really good story here, but it is easy to get lost in the details here. I am not a litter expert, but I found the way the argument was laid out very difficult to follow. I don’t have the time or energy to construct that for you and I think you and your co-authors need to spend more time structuring and editing the paper and the arguments and discussion pieces. The pieces are there, but it is not a coherent enough story yet, particularly in the discussion section. For example in the discussion you should cover in a more systematic way the strengths and weaknesses of the SM and MBM, how different are their predictions, how they could be improved. Then this should be followed by a section on key knowledge gaps, then a section on management implications. Much of this is there, but not presented in a way that is easy for the reader to follow or digest. Also, I think that the recommendation that “litter must be managed” is a bit of an over step of the results of this study, given you don’t adequately address the hydrological (soil moisture, erosion) implications of litter stocks nor the impacts of wildfire.
Specific comments:
Title: Here and throughout you say litter stocks and accumulation rates, but I see only stocks (Mg ha-1) presented and not accumulation rates (Mg ha-1 y-1)
Abstract
Line 23: maybe “measure” better than “quantify” … then “the co-related variables OF litter rainfall interception and fire regime”
Line 27: This sentence does not make sense. Please re-word.
Line 28: “employed to deliver” suggest reword to “used to predict”
Line 30-31: awkward wording… also what is the % is it r2? Were all p=0.0001?
Line 31-32 MBM “provided” a more physically based accounting… (counts does not make sense here)
Line 33-35: This is a run on sentence. Please revise into several shorter sentences.
Introduction
Line 44: This is a poor introductory sentence. Should talk about litter in this first sentence.
Line 47: I think would be better to replace “coarse” with “wood” litter and remove “made primarily of fine woody debris”.
Line 55: Intercepted rainfall does not contribute to soil moisture renewal, aquifer recharge or streamflow generation… There is no reference here and I don’t believe this is true. Litter slows down infiltration rate (and infiltration excess overland flow) which reduces erosion. Not all intercepted water is evaporated.
Line 64: Clarify what you mean by “retained rainfall limits interception of the following rain”
Line 69: Accumulation of litter is the main fuel for sustaining wildfires… please add a reference to this statement. Also, do you mean main fuel for ground wildfires. I think that wood biomass would be the main fuel in crown fires… although I’m not an expert in that field.
Line 89: remove the word “express” or change to “typically examine decay rates..”.
Line 90: This sentence “Other litterbag studies…” is awkward given the sentence above. Perhaps say most litterbag studies… in the sentence before and start the next sentence with “However, some litterbag studies have examined…”
Line 97: By composition do you mean leaves vs woody debris? If yes, please state this explicitly or explain what you mean by composition.
Line 99: Perhaps start this sentence with “For example, in the Appalachian forests in the USA…”
Line 105: Reword this awkward sentence.
Line 106: What do you mean by “co-related variables”… perhaps change to “quantify litter rainfall interception and fire regime in 133 northern forest plantations of Mexico”… however do you actually do this?
Line 110-113: This hypothesis is poorly written. Also, that the fitted models explains large variability is not a hypothesis. The last part of this run on sentence (line 113-114) is confusing.
Materials and Methods
Line 128: Suggest changing to “Tree planting in these areas..”
Line 131: change “places of” to “locations within”
Line 149: Please list the ages or age range of the chronosequence.
Line 150-151: This statement is confusing please clarify.
Line 153-154: Here and elsewhere you switch you using “I” and I suggest consistently using passive voice instead (consistently). This statement needs to be re-worded… Perhaps something like, “Additional measurements were made in 2014 in 9 plots…”
Line 157: Reword this sentence.
Line 159: Remove “I” start sentence with “Forest inventory…”
Line 165: Remove “I” start sentence with “Two years were added …”. Also, do you have a reference for this? Does two years seem reasonable for tree to reach 1.3m?
Line 168: Holorganic? Do you mean fibric and hemic layers?
Line 178: Both coarse and fine woody debris? This sentence is written awkwardly do you mean “… was included to determine total litter mass”?
Line 179: I’m confused… there was a constant distance between neighboring trees? So this is the planting density or actual density measured? Reword this sentence.
Line 182-200: This paragraph could be better organized. What is needed here is just a couple of sentences with overview of the modelling approach. Leave details for specific sections below.
Line 184-186: It is unclear why the PCA and multivariate analysis was done. If it is important list only the method here and make a section in the results for it. I’m not sure it adds to the paper.
Line 189-190: suggest change to “with the data from the nine plots re-measured in 2014 used for model validation”.
Line 191: List the measured quadrat variables. Also, you need a reference for program for stepwise procedure.
Line 193: Do you mean aboveground biomass (AGB)… not sure why you have biomass density
Line 206: change to “a correction factor of 1.5 determined from trees where both Db and DBH measured. The use of the parentheses is a bit confusing for both Db and DBH. Perhaps just choose one and stick with it or make sure to write after in parenthesis if it is Db or DBH if you use both.
Line 219: This is a bit confusing as you are referencing multiple equations with one equation number. May need to check journal format guidelines here.
Line 221: betas are “model” not “statistical” parameters
Line 230: This is not a site productivity indicator, but really a dryness index. I think it is misleading to call it also a site productivity or site index.
Line 247: It is confusing to present the equations in this way (see comment on equation above) particularly the use of the “therefore” symbol.
Results
Line 323-324: These are some large SDs! What caused that?
Table 1: You should consider increasing the number of significant figures in the table, particularly for values < 10.
Figure 3: Need to add to caption that the values at the end of the line indicate the site index at 15 years of age for xxx scenario. Perhaps also add a vertical line at 15 years.
Table 2: Check table format. Suggest repeating bold headers for MBM section. Also below table indicate what R2 and Sx are.
Discussion (see also general comments)
Line 512-530: Some of this material in this paragraph might be better suited to the introduction.
Line 550-556: Temperature is an extremely important driver of decomposition which should be discussed here.
Line 571: Yes it is hard to quantify!
Line 580: Yes, requires further study… ground vegetation coverage, needles vs broadleaf litter.
Line 588-593: Yes, lots of uncertainty here.
Line 604-606: This seems a bit premature to be recommending this give the uncertainties you list above.
Author Response
DEAR REVIEWER 3;
I APPRECIATE YOUR TIME AND CONSTRUCTIVE COMMENTS TO IMPROVE THE TECHNICAL CONTENT OF THIS REPORT. MY RESPONSE TO YOUR COMMENTS IS EMBEDDED IN YOUR TEXT IN CAPITAL LETTERS.
Summary
This paper uses a statistical and mass balance model to predict litter stocks in planted forests in northern Mexico. Litter on the forest floor plays an extremely important role in the forest in terms of carbon cycling and maintaining moisture, however is also linked to wildfire hazard. The aim of this research is to develop models to assess litter stocks, accumulation and loss rates using statistical and mass balance approaches. Their results showed that the statistical model provided better predictions. Their results also show that high productivity 40 year old stands accumulate sufficient litter that can shift the forest to a new fire regime with a risk for high severity forest wildfires. CORRECT! This research is of significance to the field, capturing a process that has not been sufficiently studied and has great implications for forest ecology. CORRECT! However, there are many instances of poor organization and grammar that the manuscript, and thus, in its current form is not ready for publication.
General comment:
I really like the use of the statistical and mass balance model approach in this paper, and I think there is a really good story here, but it is easy to get lost in the details here. THANKS! I am not a litter expert, but I found the way the argument was laid out very difficult to follow. I don’t have the time or energy to construct that for you and I think you and your co-authors need to spend more time structuring and editing the paper and the arguments and discussion pieces. The pieces are there, but it is not a coherent enough story yet, particularly in the discussion section. For example in the discussion you should cover in a more systematic way the strengths and weaknesses of the SM and MBM, how different are their predictions, how they could be improved. Then this should be followed by a section on key knowledge gaps, then a section on management implications. Much of this is there, but not presented in a way that is easy for the reader to follow or digest. Also, I think that the recommendation that “litter must be managed” is a bit of an over step of the results of this study, given you don’t adequately address the hydrological (soil moisture, erosion) implications of litter stocks nor the impacts of wildfire.
Specific comments:
Title: Here and throughout you say litter stocks and accumulation rates, but I see only stocks (Mg ha-1) presented and not accumulation rates (Mg ha-1 y-1). ACCUMULATION RATES HAS BEEN DELETED FROM THE TITLE, ABSTRACT AND MOST OF THE TEXT. HOWEVER A DERIVATIVE OF ANY OF THE TWO MODELS OVER THE DERIVATIVE OVER TIME WOULD RESULT IN ACCUMULATION, INPUTS AND LOSSES OF LITTER.
Abstract
Line 23: maybe “measure” better than “quantify” … then “the co-related variables OF litter rainfall interception and fire regime” I AM SO SORRY BUT BUT RAINFALL INTERCEPTION AND FIRE REGIME WERE NOT MEASURED BUT ASSESSED BY THE MODEL
Line 27: This sentence does not make sense. Please re-word. IT READS NOW: THE PROBABILITY OF THE F-FISHER STATISTIC TESTED THE NUL HYPOTHESIS OF MODEL SIGNIFICANCE.
Line 28: “employed to deliver” suggest reword to “used to predict”OK, CHANGED.
Line 30-31: awkward wording… also what is the % is it r2? Were all p=0.0001? IT IS THE R2 AS IT IS NOW WRITTEN IN THE TEXT.
Line 31-32 MBM “provided” a more physically based accounting… (counts does not make sense here). IT READS NOW: Results showed the SMLS model predicted and validated (p=0.0001; r2=0.82 and p=0.0001; r2=0.79) slightly better than the MBMLS model (p=0.0001; r2=0.32 and p=0.0001; r2=0.66) but the later followed tendencies of World datasets, counts for inputs, stocks, and losses from all processes and revealed decomposition loss may explain ≈ 40% of the total LS variance.
Line 33-35: This is a run on sentence. Please revise into several shorter sentences. IT READS NOW: SMLS predicted forest plantations growing in high productivity 40-yr old stands accumulate LS > 30 Mg ha-1 intercepting ≈15% of the annual rainfall and shifting to the new high-severity wildfire regime.
Introduction
Line 44: This is a poor introductory sentence. Should talk about litter in this first sentence. FIRST SENTENCE READS NOW: The soil organic litter compartment of forests consists of fine litter composed mainly of foliar materials and wood litter (Hairiah et al., 2001).
Line 47: I think would be better to replace “coarse” with “wood” litter and remove “made primarily of fine woody debris”. SEE BEFORE THANKS.
Line 55: Intercepted rainfall does not contribute to soil moisture renewal, aquifer recharge or streamflow generation… There is no reference here and I don’t believe this is true. Litter slows down infiltration rate (and infiltration excess overland flow) which reduces erosion. Not all intercepted water is evaporated. THE IDEA HAS BEEN CONSIDERABLE MODIFIED, IT READS NOW: The former is conventionally considered a loss of water from the hydrological cycle as intercepted rainfall seems to contribute little to soil moisture renewal, aquifer recharge or streamflow generation but to maintain moisture and to reduce soil erosion (Alanís et al., 2000).
Line 64: Clarify what you mean by “retained rainfall limits interception of the following rain” IT READS NOW: In general, θL usually exceeds by far IL as only part of θL is IL, but both are a function of the mass of litter stocks and θL may limit interception of the following rain (Helvey and Patric, 1965; Dunkerley, 2015).
Line 69: Accumulation of litter is the main fuel for sustaining wildfires… please add a reference to this statement. A REFERENCE HAS BEEN ADDED AND MODIFIED AS GROUND WILDFIRES. Also, do you mean main fuel for ground wildfires. I think that wood biomass would be the main fuel in crown fires… although I’m not an expert in that field. BOTH SHOULD COMPOSE THE MAIN FUELS AS ONE SUSTAINS THE OTHER AND SO ON. AND IT READS NOW: The accumulation of litter atop forest soils represents one of the main fuel sources for sustaining ground wildfires (Shlisky et al., 2007).
Line 89: remove the word “express” or change to “typically examine decay rates..”. OK, THANKS CHANGED.
Line 90: This sentence “Other litterbag studies…” is awkward given the sentence above. Perhaps say most litterbag studies… CHANGED in the sentence before and start the next sentence with “However, some litterbag studies have examined…” CHANGED, THANKS.
Line 97: By composition do you mean leaves vs woody debris? If yes, please state this explicitly or explain what you mean by composition. BOTH LEAVES vs WOODY DEBRIS AND TYPE OF LITTER, AS MANY OTHERS.
Line 99: Perhaps start this sentence with “For example, in the Appalachian forests in the USA…” IT READS NOW: For example, on average, in the Appalachian forests of USA, high-severity wildfires may reduce the mass of litter stocks from 180 Mg ha-1 to 70 Mg ha-1 (Vose and Swank, 1993) and worldwide statistics show high-severity, large-scale forest wildfires can burn a mean of 140 Mg ha-1 of the total litter stocks (van der Werf et al., 2004; Giglio et al., 2006).
Line 105: Reword this awkward sentence. IT READS NOW: IN LIGHT OF THIS BRIEF LITERATURE REVIEW.
Line 106: What do you mean by “co-related variables”… perhaps change to “quantify litter rainfall interception and fire regime in 133 northern forest plantations of Mexico”… however do you actually do this? WE CALCULATE LITTER INTERCEPTION AND FIRE REGIMES WITH MODELED LITTER STOCKS. THE SENTENCE HAS BEEN CHANGED TO EVALUATE INSTEAD OF QUANTIFY.
Line 110-113: This hypothesis is poorly written. Also, that the fitted models explains large variability is not a hypothesis. The last part of this run on sentence (line 113-114) is confusing. IT READS NOW: The central hypothesis of this study was that fitted models would statistically explain at least part of the large spatial variability expected in the mass of litter stocks measured in three different forests (Durango, Coahuila and Nuevo Leon). Measured forest plantations have different basal area or stand density at the time of planting and at the time of measurements. Plantations with different pine species may also present different litter fall rates, decomposition factors, among many other traits.
Materials and Methods
Line 128: Suggest changing to “Tree planting in these areas..” CHANGED, THANKS.
Line 131: change “places of” to “locations within” CHANGED, THANKS.
Line 149: Please list the ages or age range of the chronosequence. ADDED RANGE AND AVERAGE.
Line 150-151: This statement is confusing please clarify. IT READS NOW: A representative set of plantation was chosen with different dimensions assuming planting times were different.
Line 153-154: Here and elsewhere you switch you using “I” and I suggest consistently using passive voice instead (consistently). PASSIVE VOICE IS NOW USED. This statement needs to be re-worded… Perhaps something like, “Additional measurements were made in 2014 in 9 plots…” IT READS NOW: Additional measurements were made in 2014 in 9 previously-sampled plots in Durango with the aim to validate models.
Line 157: Reword this sentence. IT READS NOW: Forestry parameters serve as exogenous variables to predict litter stocks using either empirical or mass balance budget models.
Line 159: Remove “I” start sentence with “Forest inventory…” NOW IN PASSIVE VOICE.
Line 165: Remove “I” start sentence with “Two years were added …”. Also, do you have a reference for this? Does two years seem reasonable for tree to reach 1.3m? IT READS NOW: Two years were added to the final ring count to consider the time needed for the trees to reach 1.3 m after planting. This seems reasonable according to the height growth and yield models developed by Corral-Rivas and Návar-Cháidez (2005).
Line 168: Holorganic? Do you mean fibric and hemic layers? YES FEMIC AND HEMIC. SORRY.
Line 178: Both coarse and fine woody debris? YES, BOTH, This sentence is written awkwardly do you mean “… was included to determine total litter mass”? IT READS NOW: The mass of both coarse and fine woody debris was included to determine total litter mass.
Line 179: I’m confused… there was a constant distance between neighboring trees? So this is the planting density or actual density measured? Reword this sentence. IT IS ASSUMED INITIAL PLANTING DENSITY BY MEASURING THE DISTANCE OF SEVERAL NEIGHBORING TREE. IT READS NOW: The initial planting density was evaluated from the distance of the nearest neighbor trees for several trees with similar dimensions and the final planting density was calculated as the number of tress per ha at the time of measurement.
Line 182-200: This paragraph could be better organized. What is needed here is just a couple of sentences with overview of the modelling approach. Leave details for specific sections below. OK. IT READS NOW: Two different models were developed to evaluate the mass of litter stocks: a statistical model using stepwise procedures, SMLS, and a mass balance budget model using litter inputs (gains), changes in storage (present litter stock) and calibrating it for litter outputs or losses, MBMLS.
Line 184-186: It is unclear why the PCA and multivariate analysis was done. If it is important list only the method here and make a section in the results for it. I’m not sure it adds to the paper. DELETED FROM THE TEXT.
Line 189-190: suggest change to “with the data from the nine plots re-measured in 2014 used for model validation”. CHANGED, THANKS, IT READS NOW: Litter stock models were developed only using the data collected during 2002-2003, with the data from the nine plots re-measured in 2014 used for model validation.
Line 191: List the measured quadrat variables. Also, you need a reference for program for stepwise procedure. DONE. THANKS.
Line 193: Do you mean aboveground biomass (AGB)… not sure why you have biomass density. YES, I MEAN AGB.
Line 206: change to “a correction factor of 1.5 determined from trees where both Db and DBH measured. The use of the parentheses is a bit confusing for both Db and DBH. Perhaps just choose one and stick with it or make sure to write after in parenthesis if it is Db or DBH if you use both. SORRY, I WILL TRY TO USE BOTH.
Line 219: This is a bit confusing as you are referencing multiple equations with one equation number. May need to check journal format guidelines here. OK. THANKS. I LEFT BOTH EQUATIONS AS THE LATER IS DERIVED FROM THE FORMER.
Line 221: betas are “model” not “statistical” parameters. OK. THANKS.
Line 230: This is not a site productivity indicator, but really a dryness index. I think it is misleading to call it also a site productivity or site index. OK. AND DRYNESS IS RELATED TO PRODUCTIVITY VERY CLOSELY. I USED BOTH TERMS, ADDING YOUR COMMENT ON DRYNESS. THANKS-
Line 247: It is confusing to present the equations in this way (see comment on equation above) particularly the use of the “therefore” symbol. OK. THANKS. THE EQUATION WAS CHANGED BUT IT REMAINS ESSENTIALLY THE SAME
Results
Line 323-324: These are some large SDs! What caused that? THE INITIAL PLANTING DENSITY WAS HIGHLY VARIABLE; FROM 10,000 TO 400 SEEDLINGS PER HA WERE PLANTED.
Table 1: You should consider increasing the number of significant figures in the table, particularly for values < 10. NUMBERS ARE RUNDED OFF. THANKS.
Figure 3: Need to add to caption that the values at the end of the line indicate the site index at 15 years of age for xxx scenario. Perhaps also add a vertical line at 15 years. OK. THANKS. DONE.
Table 2: Check table format. Suggest repeating bold headers for MBM section. Also below table indicate what R2 and Sx are. BOLD HEADERS AND DESCRIPTION OF R2 AND Sx.
Discussion (see also general comments)
Line 512-530: Some of this material in this paragraph might be better suited to the introduction. GREAT THANKS. PART OF THIS WAS MOVED TO THE INTRODUCTION IN THE SECTION OF WILDFIRES.
Line 550-556: Temperature is an extremely important driver of decomposition which should be discussed here. THE MEAN ANNUAL TEMPERATURE HAS BEEN DESCRIBED WITHIN THIS TEXT TO MAKE THE POINT. THANKS.
Line 571: Yes it is hard to quantify! THANKS FOR THE REASSURANCE OF THIS STATEMENT.
Line 580: Yes, requires further study… ground vegetation coverage, needles vs broadleaf litter. THANKS FOR THE TIP.
Line 588-593: Yes, lots of uncertainty here. READY! THANKS FOR THE TIP.
Line 604-606: This seems a bit premature to be recommending this give the uncertainties you list above. BUT WE MEASURED LITTER STOCS > 30 Mg HA-1 AND MODELS PREDICT WELL THIS MASS AS WELL.
Reviewer 4 Report
This paper modeled the litter stocks and accumulation rates, which was a work of great significance. A well simulated model can estimate the future litter stocks and accumulation rates of plantation forests through some simple and easy measured indicators. However, there are still some serious problems in the article. I don't know whether the data 20 years ago are still suitable for the current climate and forest conditions? As the description in your results, the planting density has decreased by 36% after 16 years; Secondly, the language of the article is slightly obscure and difficult to understand. In addition, there are many small mistakes. The authors need to read and modify them carefully. I list some below; Finally, I'm not sure, because looking at the regression model of tables 2 and 3, the simulation effect is not very good, such as Eq. 1, 2, 6 and 8 in table 2 and result 3.3. I don't think this result is enough to support the author's conclusion. Here are the specific issues
line110-114 The last sentence of the Introduction, how does the last part of the sentence (the large diversity of planted pine species in these forests) relate to this sentence.
Line151-153 Each site has only one quadrat, can it represent the situation of one site?
Line168 what does OL, OF and OH mean?
Regarding the layout of the figures and tables, why are they inserted in the text, but not in the position of the first reference?
Line 211-214 After the equation is adjusted in this way, will there be no problem in BA estimation of other locations? Because you said that before the adjustment, only the BA of Durango plantations was underestimated.
Line 221 B0, B1, and B2 maybe ?0,?1 and ?2?Because the letter in your equation (1) is really B, but here is ?,Please don't make a mistake here。And line 249 SI15 is SI?
Figure 2 there is something wrong with this figure, the notes are garbled
Line364 (r2 ≤ 0.24; p=0.0025), is this a correct statement?
Line407-411 The full points in the text disappeared.
Line 451-452 Is this a straight-line relationship? And where is Figure 6.
References: The references are too old. The references of the latest year are still in 2015, and there are no updated references.
Author Response
Dear Reviewer 4;
Comments:
Thanks for STRESSING the significance of this STATE of the ART report on MODELING LITTER STOCKS.
Please remember that data to validate the models was collected in 2014 and models show their strengths also with this data source. Remember that models also include the P/Et parameter that should the climate be changing these parameters would also change.
The language was checked by several well known external companies but I recognize there still some need for improvement. I also appreciate your comments on how to improve the language.
Finally, please look at Figure 5, in both Figures (5a and 5b) the MBM model shows how predictions match well the measured averages for local, regional and world forests stressing its validation.
SPECIFIC COMMENTS
Line 110-114. The sentence was expanded to describe different species present different litter fall rates and litter output, more specifically decomposition factors.
Line 151-153: We measured litter stocks and forest attributes in 133 planted forests: 47, 46 and 40 in Durango, Nuevo Leon and Coahula, respectively. In each quadrat we sampled three three (3) places.
L 168. OL=organic litter; OF = fumic and OH = humic litter. Described now in text.
Now Figures and Tables are positioned in the place where they are first referenced. Thanks
Line 211-214. The initial fitting of the model, yes it underestimated. THAT IS WHY THE EXPLANATION OF FORCING THE MODEL TO REACH 50 m2 ha-1 at 80 years of age; to be consistent with other regional BA models.
Line 221. Yes, Thanks B is now in greek letters. SI was estimated at 15 years of age, SI15, as the age of these plantations does not esceed 45 years.
Fig 2. The garbling problem is from the Graph System. I will arrange it by saving it in a different format. Thanks
Line 451-452. Yes, it is a linear interpolation as there is no local information on litter interception of rainfall.
REFERENCES. Too old. Hope you could provide with updated references. Modeling litter stocks is absent in the Scientific Literature.
Round 2
Reviewer 2 Report
This manuscript provides important aspects to guide forest management based on statistical relationships between litter stocks and other factors. This approach has the potential to make a better decision of forest management plan. Despite such motivations and relevance of study, the quality of writing still has to be improved substantially. There are many places where the sentences are not connected well to the next sentence, leading to difficult to review.
For example:
L40: this first paragraph is important to expand the purpose of this study – ‘Important roles of organic litters on forest floor’; however, the authors developed the idea as the following orders: 1) components of soil organic litter; 2) C and N are the main components of litter; 3) large sizes of forest covers contains huge AGB; 4) litter is a major sink and flux of carbon and nitrogen and holds as much biomass as the AGB compartment.
If the authors read carefully this, the connectivity between ideas is not cohesive.
I suggest 1) soil organic litters are the key in the forest floor as its important role in water cycling in ecosystems; 2) details of roles in the water cycle and its potential impact on wildfire; 3) litters are strongly correlated to the amount of AGB, so how C and N-related indicators in litters affect the amount of AGB needs to be investigated for a better understanding of ‘what the authors want to address’.
And the authors need to develop details (I think components of statistical models to justify why those components were used?).
Otherwise, the readers are likely to be distracted while they read this manuscript.
Specific comments:
L15: I suggest ‘Litter (LS)’; maybe ‘Litter is the organic material in which locates in the top of A soil horizon, playing key ecological roles in forests’; remove ‘excluding roots’
L15-18: the authors directly got into the main point without addressing knowledge gap that the study wants to fill; where is the knowledge gap?
L23-24: this statement is not necessary to mention in the abstract
L26-27: ‘SMLS model outperformed (explained 82% of variances) than MBMLS model’, what is the meaning of predicted and validated here? Is that meaning model’s R-square and in-sample prediction’s R-square? It needs to be clear
L32: please remove ‘in the meantime’
L42: please add ‘such as regulating water cycles and fuels of wildfire’
L42-43: not sure ‘Specifically, litter contains significant amounts of several major bio-geochemicals including carbon and nitrogen’ is relevant here
L43-45: those sentences need to be reorganized or a better framed
L50-51: ‘up to 70%’
L55: The authors used the term of ‘loss from the hydrological cycle’, but it should be water redistributions
L61: Two major components are considered as the key element for litter-mediated changes in water cycling: water holding capacity and depth of rainfall intercept
L63-65: I could not follow this statement.
While water holding capacity is often larger than depth of rainfall intercept, those two factors use to determine of the mass of litter stocks… Or
While water holding capacity is often larger than depth of rainfall intercept, water holding capacity could limit interception of the following rain?
This transition is not really smooth; the authors need to find a smooth way to transit from ‘each component’s roles in water cycle’ to ‘the thing of mass balances’ to define the knowledge gap that this study wants to fill
Author Response
Dear Reviewers 2 & 3;
I again greatly appreciate your help through observations, details, comments on the manuscript titled ‘Modeling Litter Stocks in Planted Forests of Northern Mexico’ to improve the technical content as well as the English Grammar and Style. RESPONSE TO CONCERNES ARE EMBEDDED IN CAPITAL LETTER THROUGHOUT THE MANUSCRIPT.
Kind Regards
Sincerely
Jose Navar, PhD
REVIEWER 2
This manuscript provides important aspects to guide forest management based on statistical relationships between litter stocks and other factors. THANKS. This approach has the potential to make a better decision of forest management plan. THAT IS THE IDEA. Despite such motivations and relevance of study, the quality of writing still has to be improved substantially. OK. There are many places where the sentences are not connected well to the next sentence, leading to difficult to review. OK. I AM CONDUCTING A THROUGHOUT REVIEWEW AND DO THE REVISIONS YOU RECOMMEND WITH THE AIM TO FIX THESE PROBLEMS.
For example:
L40: this first paragraph is important to expand the purpose of this study – ‘Important roles of organic litters on forest floor’; however, the authors developed the idea as the following orders: 1) components of soil organic litter; 2) C and N are the main components of litter; 3) large sizes of forest covers contains huge AGB; 4) litter is a major sink and flux of carbon and nitrogen and holds as much biomass as the AGB compartment. IT READS NOW: Litter plays key roles in the ecology of forests hence the importance of measuring, developing and validating models to predict LS over time of forest growth. These mathematical techniques would help to objectively propose silvicultural practices that aim to optimize ecological functions while at the same time improving the soil water balance to increase productivity and reduce the risk of high-severity wildfires.
If the authors read carefully this, the connectivity between ideas is not cohesive. OK. THANKS FOR THE OBSERVATION, REMEMBER THAT MAJOR ASSUMPTION IS THAT READERS OF SCIENTIFIC DOCUMENTS UNDERSTAND THESE CONCEPTS AND HENCE NO NEED TO EXPAND BUT TO FOLLOW TO THE NEXT IDEA.
I suggest 1) soil organic litters are the key in the forest floor as its important role in water cycling in ecosystems; 2) details of roles in the water cycle and its potential impact on wildfire; 3) litters are strongly correlated to the amount of AGB, so how C and N-related indicators in litters affect the amount of AGB needs to be investigated for a better understanding of THE WATER BUDGET AND THE RISK OF HIGH-SEVERITY WILDFIRES‘what the authors want to address’.
And the authors need to develop details (I think components of statistical models to justify why those components were used?). IT READS NOW: However predicting models cannot be found in a brief scientific literature review and common constant allocation factors had been conventionally employed in LS assessments (Vitousek et al., 2004). Forest growth and yield models aid in the evaluation of timber growth (Vanclay, 2004) and biomass expansion factors, BEF, (Brown. 1997) can aid in the evaluation of AGB as the major input of LS by developing empirical equations. Whole stand timber growth and yield models are based on evaluating stand variables such as basal area, BA, productivity indicators such as site index (Carmean et al., 1989) and age of forests (Peng et al., 2001). These variables can be used as input exogenous predictors to more complex statistical and budget techniques of LS evaluations. Once LS had been predicted, its role in the local hydrology as well as on the wildfire regime can be used to recommend forest management practices aiming to improve the local water balance and to reduce the risk of high severity wildfires.
Otherwise, the readers are likely to be distracted while they read this manuscript.
Specific comments:
L15: I suggest ‘Litter (LS)’; maybe ‘Litter is the organic material in which locates in the top of A soil horizon, playing key ecological roles in forests’; remove ‘excluding roots’ IT READS NOW: Litter, LS, is the organic material in which locates in the top A soil horizon, playing key ecological roles in forests.
L15-18: the authors directly got into the main point without addressing knowledge gap that the study wants to fill; where is the knowledge gap? IT READS NOW: Models, in contrast to common allocation factors, must be used in LS assessments as they are currently absent in the scientific literature.
L23-24: this statement is not necessary to mention in the abstract. OK. THE SENTENCE HAS BEEN DELETED.
L26-27: ‘SMLS model outperformed (explained 82% of variances) than MBMLS model’, what is the meaning of predicted and validated here? Is that meaning model’s R-square and in-sample prediction’s R-square? It needs to be clear
L32: please remove ‘in the meantime’ OK. IT HAS BEEN DELETED. BUT THE SENTENCE DOES NOT MAKE SENSE ANY MORE. SO IT WAS MODIFIED AS FOLLOWS: SMLS is preliminarily recommended for LS management and predicts the need of management in forest plantations (> 40-yr old) to reduce rainfall interception as well as the risk of high-severity wildfires.
L42: please add ‘such as regulating water cycles and fuels of wildfire’ OK. THIS STATEMENT HAS BEEN ADDED. IT READS NOW: Litter plays key roles in the ecology of forests such as regulating water cycles and fuels of wildfire. Hence the need of measuring, developing and validating models to predict LS over time of forest growth. These mathematical techniques would help to objectively propose silvicultural practices that aim to optimize ecological functions while at the same time improving the soil water balance to increase productivity and reduce the risk of high-severity wildfires.
L42-43: not sure ‘Specifically, litter contains significant amounts of several major bio-geochemicals including carbon and nitrogen’ is relevant here. OK. THE FULL SENENCE HAS BEEN DELETED. THANKS
L43-45: those sentences need to be reorganized or a better framed. AS YOU RECOMMENDED BEFORE, RECALLING:
I suggest 1) soil organic litters are the key in the forest floor as its important role in water cycling in ecosystems; 2) details of roles in the water cycle and its potential impact on wildfire; 3) litters are strongly correlated to the amount of AGB, so how C and N-related indicators in litters affect the amount of AGB needs to be investigated for a better understanding of THE WATER BUDGET AND THE RISK OF HIGH-SEVERITY WILDFIRES‘what the authors want to address’.
L50-51: ‘up to 70%’ SORRY, BUT IT IS WHAT THE LITERATURE SAYS ABOUT LITTER RAINFALL INTERCEPTION.
L55: The authors used the term of ‘loss from the hydrological cycle’, but it should be water redistributions. OK. IT HAS BEEN REPLACED. THANKS.
L61: Two major components are considered as the key element for litter-mediated changes in water cycling: water holding capacity and depth of rainfall intercept. OK. THANKS. NOW IT READS: Two major components are considered as the key element for litter-mediated changes in water cycling: water holding capacity, θL, and depth of rainfall intercep, IL, have been recognized as important elements in litter interception (Dunkerley, 2015).
L63-65: I could not follow this statement. IT READS NOW: In most rains, θL exceeds IL as only part of θL may become IL, but both are a function of litter stocks. SORRY, I DID NOT SEE YOUR RECOMMENDATIONS BELOW
While water holding capacity is often larger than depth of rainfall intercept, those two factors use to determine of the mass of litter stocks… Or
While water holding capacity is often larger than depth of rainfall intercept, water holding capacity could limit interception of the following rain? IT READS NOW: While θL capacity is often larger than depth of rainfall IL, θL could limit IL,of the following rain; those two factors use to determine litter stocks (Helvey and Patric, 1965; Dunkerley, 2015).
This transition is not really smooth; the authors need to find a smooth way to transit from ‘each component’s roles in water cycle’ to ‘the thing of mass balances’ to define the knowledge gap that this study wants to fill. THANKS FOR THE OBSERVATION, COULD YOU PLEASE HELP ME TO ADDRESS THIS ISSUE BETTER WITH SOME RECOMMENDATIONS.
Reviewer 3 Report
Please check English grammar and style throughout.
Author Response
Dear Reviewers 2 & 3;
I again greatly appreciate your help through observations, details, comments on the manuscript titled ‘Modeling Litter Stocks in Planted Forests of Northern Mexico’ to improve the technical content as well as the English Grammar and Style. RESPONSE TO CONCERNES ARE EMBEDDED IN CAPITAL LETTER THROUGHOUT THE MANUSCRIPT.
Kind Regards
Sincerely
Jose Navar, PhD
REVIEWER 3
Please check English grammar and style throughout. OK. THANKS. I DID THIS BY USING THE ENGLISH GRAMMAR OF MICRO SOFT WORD.